# Spark ablation metal nanoparticles and coating on TiO$_2$ in the aerosol phase

Benjamin Gfeller[1], Mariia Becker[1], Adrian D. Aebi[1], Nicolas Bukowiecki[1], Marcus Wyss[2], Markus Kalberer[1]

[1]Department of Environmental Sciences, University of Basel, Basel, 4056, Switzerland
[2]Swiss Nanoscience Institute, University of Basel, Basel, 4056, Switzerland

*Correspondence to*: Markus Kalberer (markus.kalberer@unibas.ch)

**Abstract.** Generation and characterisation of metal nanoparticles (NPs) in the aerosol phase gained attention in recent years due to their significant potential in applications as diverse as catalysis, electronics or energy storage. Despite the high interest in NPs, it remains challenging to obtain detailed quantitative information with conventional aerosol analysis instruments on size, number concentration, coagulation behaviour and morphologies, which are key to understand their properties. In this study we generated NPs from four metals, Au, Pt, Cu and Ni, via spark ablation in the aerosol phase, which allows to produce NPs as small as 1 nm in high quantities and purity. Particles were characterised with transmission electron microscopy (TEM), scanning transmission electron microscopy (STEM) and energy dispersive X-ray spectroscopy (EDX) as well as online aerosol particle size distribution measurement techniques. Particle size modes for the four metals ranged between 3 nm and 5 nm right after generation. Differences in number and size of particles generated can be rationalised with thermodynamic properties of the metals such as melting point combined with their oxidative properties. The four metal NPs were also coagulated with larger TiO$_2$ NPs of about 120 nm size and the metal surface coverage of the TiO$_2$ particles was characterised with electron microscopy and EDX spectroscopy. This detailed characterisation of NPs mixtures will be essential for a fundamental understanding of spark ablation generated particles and their applications for material sciences.

## 1 Introduction

Nanoparticles (NPs) have long been discovered to have unique physical and chemical properties such as their changed hardness, elastic modulus, adhesion force(Guo et al., 2013; Khan et al., 2019) as well as electronic, optical and catalytic properties(Baig et al., 2021; Cuenya, 2010; Jennings and Strouse, 2007; Terna et al., 2021) compared to bulk material. Understanding the working principles on the nano-scale is crucial for new and emerging applications in energy harvesting(Ulmer et al., 2019), medicine(Murthy, 2007), electronics(He et al., 2020) etc. to be improved.

There is a wide range of generation methods for metal NPs ranging from chemical methods such as chemical reduction(Gudikandula and Charya Maringanti, 2016) and sol-gel methods(Lu and Jagannathan, 2002) to physical methods like laser or spark ablation (Nagarajan, 2008; Schwyn et al., 1988; Ullmann et al., 2002). Chemical approaches which usually produce NPs in batch processes are well established often yielding reproducible output within narrow particle size

distributions.(Daruich De Souza et al., 2019; Jamkhande et al., 2019; Zhang et al., 2010) However, the purity of the generated particles can be compromised due to solvent residues or the toxicity of reagents can pose challenges.(Jamkhande et al., 2019; Zhang et al., 2010) In comparison, in physical generation methods, particles are often generated in an inert gas, allowing for a continuous particle production usually yielding particles in high purity due to the absence of liquid precursors

and solvent.(Kumari et al., 2023)

Spark discharge generation (SDG) is a physical generation method that produces a high quantity and purity of NPs. During particle generation in SDG, electrode material is evaporated in the vicinity of an electrical spark initiated via the break-down of a high voltage difference applied to the electrodes.

The discharge occurs on time scales of microseconds at temperatures of around 20'000 K (Reinmann and Akram, 1997). At

these temperatures, electrode material is evaporated and immediately mixed with a carrier gas. Therein, the electrode vapor expands rapidly and cools down with quenching rates of $10^7$ $Ks^{-1}$ resulting in the homogeneous nucleation of the evaporated electrode material to NPs with sizes well below 10 nm (Pfeiffer et al., 2015). This method is highly versatile as every conductive and solid pure element or alloy can be used for particle generation. (Schwyn et al., 1988) Compared to laser ablation, no high energy lasers are needed for SDG particle generation, making it technically a more straight forward

process. (Tabrizi et al., 2009) The generation of metal NPs via SDG has gained increasing attention in recent years (Hallberg et al., 2018; Petallidou et al., 2023; Snellman et al., 2024) due to the potential applications of NPs e.g., in catalysis(Weber et al., 1999), nanoprinting(Jung et al., 2021) and drug delivery(Murthy, 2007). However, characterisation of the produced particles in the gas phase, e.g. their quantification, is challenging.(Kangasluoma et al., 2020) Given the high diffusion coefficients of sub 10 nm particles (Hinds, 1999), sampling must occur rapidly after particle generation to minimise losses

due to diffusion to walls of the experimental setup or coagulation with other particles. Currently available techniques to determine aerosol NP number size distributions mainly use size classification via mobility analysis using differential mobility analysers (DMA) and detection via condensation particle counters (CPC) or electrometers.(Chen et al., 1998) Electrical mobility methods suffer from low charging efficiencies of particles in the low nm size range(Fuchs, 1963) resulting in low detection efficiencies. Although CPCs can detect particles to sizes down to about 2 nm (Brilke et al., 2020),

detection is challenging because particles need to get activated in a supersaturated vapour (usually butanol or water) before detection of the activated particle by light scattering and due to diffusional losses of the smallest particles. Furthermore, these methods do not allow to assess particle morphologies. High-resolution imaging techniques are required for morphological analysis of particles as small as 1 nm and electron microscopy is an alternative method to determine particle number and size distributions down to this size range.(Fissan et al., 2014; Karlsson et al., 2006)

After generation, NPs can be further manipulated in a number of ways, e.g. by depositing small NPs on the surface of larger (substrate) particles in the gas phase resulting in particles with complex chemical or physical properties. (Pfeiffer et al., 2015; Snellman et al., 2024) Metal NPs can be used to coat semiconducting particles such as $TiO_2$, MgO or $CuO_2$.(Gao et al., 2015; Hejral et al., 2013; Lopez, 2004; Molina and Hammer, 2005) Resulting structures were predicted theoretically (e.g. Molina 2005) and were shown empirically to have increased catalytic activities for reactions such as methanation (Gao

2015) or CO oxidation (Lopez 2004). Different methods are available to assess the efficiency of such coating processes. Microscopic imaging can visualise coating structures and assess them qualitatively in high resolution.(Harra et al., 2015; Pfeiffer et al., 2015) Furthermore, measurements of the aerosol number size distributions of the two composites individually (i.e. substrate NPs and coating NPs) and during the coating via mobility sizing and counting are often conducted to quantify the coating. (Backman et al., 2004; Lähde et al., 2008)

As mentioned above, particles generated via SDG are often in the size range of only a few nm, which is challenging for most conventional aerosol particle analysis techniques, such as scanning mobility particle sizing (SMPS) instruments. Electron microscopy analyses, e.g. high-resolution transmission electron microscopy (TEM) suffer from other sampling and analysis artefacts to determine total concentrations and size distributions of particles in the aerosol phase. By combining electron microscopy with aerosol techniques (i.e. SMPS), we investigate the size distribution of four SDG-generated metal NP

species (Au, Pt, Cu and Ni) and develop a robust method to determine their size distribution in the aerosol phase and the size range of primary particle for all four metals, which range from <1 nm to about 5 nm using particle circularity as a key metric.

A second aspect of this study addresses the coagulation behaviour of the four metals NPs with larger (~ 120 nm) $TiO_2$ NPs , which is also challenging to characterise and to quantify. Electron microscopy allows for counting of metal NPs coating the

larger $TiO_2$ particles but such conventional methods often suffer from poor counting statistics and therefore major uncertainties. We address the difficulty of estimating NP coating processes and present a method to estimate average coating efficiencies by quantifying the number concentrations of metals NPs which did not coagulate with $TiO_2$ particles and compare these numbers with NPs in TEM analyses when no $TiO_2$ particles are present. We illustrate that electron microscopy is well suited to characterise quantitatively coating processes of metal NPs on larger $TiO_2$ particles. Such

thorough characterisations are important for a fundamental understanding of spark ablation generated NPs and their chemical or physical properties.

**2 Experimental**

Figure 1 shows a general overview of the experimental setup used to generate metal and TiO$_2$ particles and collect them for characterisation.

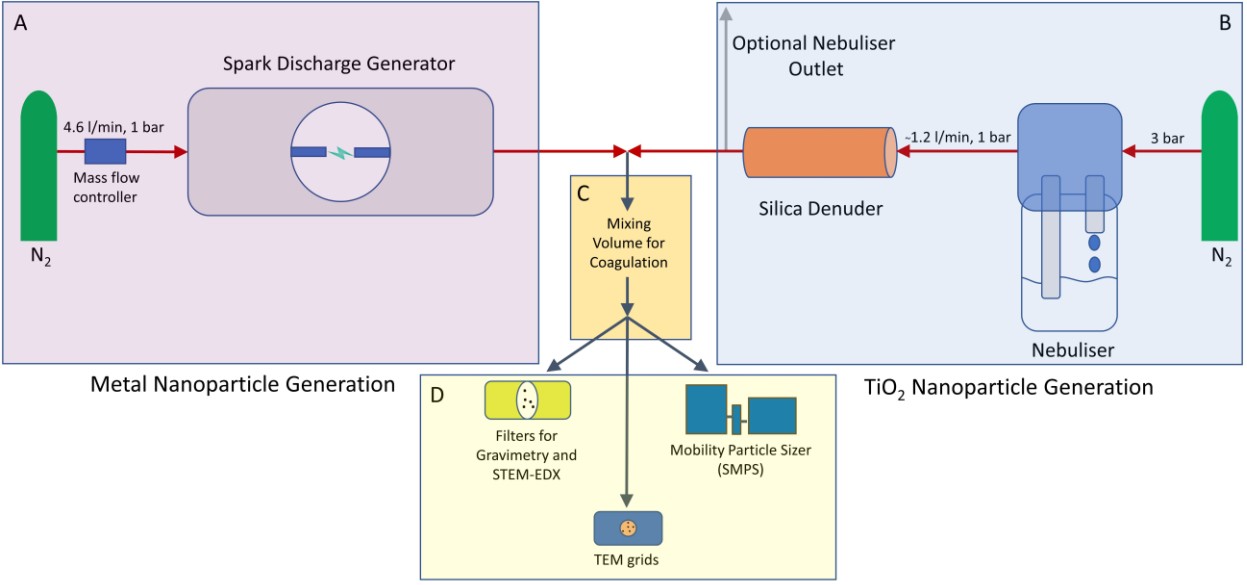

**Figure 1: Schematic of the setup used to generate (A, B), process (C) and analyse (D) metal NPs. The spark generated particles (A) are either mixed with nebulised TiO$_2$ substrate NPs (B) or, after passing a variable volume for coagulation (C), directly analysed with transmission electron microscopy (TEM), scanning transmission electron microscopy (STEM) and STEM in combination with energy dispersive X-ray spectroscopy (STEM-EDX) and by scanning mobility particle sizing (SMPS) (D).**

## 2.1 Particle generation and mixing

Metal NPs were generated with a spark discharge generator (SDG) (VSParticle, Model G1, Delft, NL) (Fig. 1, A). In a SDG, the breakdown of a high voltage applied to two electrodes causes a discharge spark. The high temperatures of the spark (up to 20'000 K) evaporate electrode material, which is quickly quenched in a continuous carrier gas flow causing the evaporated material to condense into particles consisting of the electrode material.

To ensure comparability, the discharge voltage of the instrument was set to 1 kV with a current of 5 mA in all experiments. Au, Pt, Cu and Ni nanoparticles were generated with the SDG using electrodes of the respective metal with a purity of 99.99% and a diameter of 3 mm. The SDG was run in crossflow mode, where the quenching gas N$_2$ (99.999% purity) entered the spark chamber perpendicular to the electrodes at the position of the spark. If not specified otherwise, the N$_2$ flow rate was set to 5.8 l/min. To decrease instabilities in the particle generation, the system ran for 30 min before further particle processing and collection. This equilibration process was monitored using a scanning mobility particle sizer (SMPS, DMA model 3085, CPC model 3776, TSI, Shoreview, USA).

In addition to metal nanoparticles, TiO$_2$ aerosol particles were generated, using a home-built nebuliser (Fig. 1, B) containing a 1%wt suspension of TiO$_2$ particles with a rutile and anatase mixture and a BET determined particle size <100nm (99.5% purity, Sigma Aldrich, Burlington, USA). The nebuliser inlet is pressurized with N$_2$ (99.999%) at 3 bar. MiliQ® (resistance >18 MΩ) water was used for the preparation of the suspension to minimise impurities. Before use, the TiO$_2$ suspension was sonicated for 10 min and stirred continuously during operation to minimize coagulation within the suspension. The nebulised aerosol with a flowrate of 1.2 +/- 0.1 l/min passed through a cylinder filled with amorphous silica to reduce the relative humidity to < 2%. Analogously to the SDG, the particle output of the nebuliser was left to equilibrate for 30 min before the analysis. Whenever the nebuliser was in use, to maintain a constant total flow of 5.8 l/min, the flow through the SDG was reduced to 4.6 l/min.

After exiting the SDG or the nebuliser, the aerosol particles entered a variable mixing volume (Fig. 1, C) with total coagulation times of 1.3 s, 2.2 s or 26.0 s (i.e. time between exiting the spark generator (and optional mixing with the TiO$_2$ NPs) and particle deposition), respectively, to allow for mono- or bi-modal coagulation (coating). For mono-modal coagulation of metal NPs, the nebuliser was switched off. For bi-modal coagulation, the flow through the spark generator was reduced to 4.6 l/min to maintain constant coagulation times. Due to this slight change in the SDG flow rate, changes in particle characteristics cannot be excluded but such changes should be minimal as described in (Tabrizi et al., 2009). If not mentioned otherwise, conductive Tygon tubing (TSI, Shoreview, USA) was used throughout the set up.

## 2.2 Particle collection

Aerosol particles were collected on TEM grids, filters or on TiO$_2$ substrate films. For TEM, STEM and STEM-EDX particles were collected in a diffusional collection chamber on TEM grids (Quantifoil® R 1.2/1.3 on Cu or Au 200 mesh grids + 2 nm C, Grosslöbichau, Germany) for 2 h for each configuration. Teflon filters (2 μm pore size, 47 mm diameter, Pall Corporation, Port Washington, USA) were installed after the coagulation volumes to collect the aerosol particles for gravimetric analyses for 6h for all metals. Aerosol particle size distribution measurements confirmed quantitative collection of the metal and TiO$_2$ NPs.

Alternatively, a film of TiO$_2$ substrate (TiO$_2$ electrodes opaque, Solaronix, Aubonne, CH) was used and exposed to a metal aerosol flow for 6h (Au and Pt) and 12h (Cu, Ni).

## 2.3 Particle analysis

Several measurement techniques were used to characterise metal and TiO$_2$ NPs (Fig. 1, D). Particles were analysed with TEM as well as STEM and STEM-EDX (JEM-F200 cFEG, Jeol, Tokyo, Japan). For STEM, high-angle annular dark-field (HAADF) and annular bright-field (ABF) detectors were used. Au, Pt, Cu and Ni particles were measured separately or coated on TiO$_2$, in TEM mode. The coated aerosol particles were further analysed in STEM mode and STEM-EDX. The

image analysis tool ImageJ (Fiji, v.1.54.f) was used for quantification and characterization of the particles. After differentiation of the particles and the background grid via manual adjustment of the colour threshold of the images, quantitative (2D projected area and number of particles within one $\mu m^2$ of the grid) and qualitative information (morphology) were determined.

Gravimetric measurements of particles collected on Teflon filters were conducted using a high precision balance (Model XPR2, Mettler Toledo, Columbus, USA).

For lamella analysis, which provide a cross sectional cut through nano-structured samples, protective coating layers are often applied to minimize charging effects. For the STEM-EDX analysis of metal NP coated $TiO_2$ films, lamellas were prepared using a FEI Helios NanoLab 650 DualBeam (FEI, Hillsboro, USA) and a protective layer of Au or C was deposited with a sputter coater before the lamella preparation. Since the $TiO_2$ film surface is rough, a layer of about 100 nm thickness was deposited. Au or C was used to prevent the transitions in the EDX from overlapping with the material of the NPs. A Pt or C-layer was then deposited onto this protective layer using first electron-induced deposition (5 keV, 3.2 nA) of approximately 200 nm and afterwards ion-induced deposition (30 keV, 83 pA) of approximately 800 nm in a small defined rectangular area for lamellae preparations. Sample cutting and polishing were carried out with the focused ion beam at a beam energy of 30 kV and beam currents ranging from 240 pA down to 83 pA. The sample thickness was < 75 nm. The imaging of TEM lamellas was carried out with a JEOL JEM-F200 instrument operated in STEM-mode at a beam energy of 200 kV.

A scanning mobility particle sizer (SMPS, TSI, Shoreview, USA) consisting of a soft X-ray neutralizer (Model 3087), electrostatic classifier (Model 3080), DMA (Model 3085) and CPC (Model 3776) was used to monitor particle size distributions. The instrument was operated with a 1.5 l/min sample flow and a 15 l/min sheath air flow and a scanning range of 1.5 nm to 64 nm with a theoretical 50% detection efficiency diameter ($D_{50}$) of 2.5 nm of the CPC.

# 3 Results and Discussion

## 3.1 Metal nanoparticles

### 3.1.1 Particle morphologies

Au, Pt, Cu and Ni metal NPs were generated with the SDG and collected for 2 hours, approximately 1.3 s after exiting the spark generator, by diffusion onto TEM grids. They were then analysed via TEM for their morphology. Fig. 2 shows TEM micrographs of all four metals, with particle sizes ranging between < 2 nm up to > 50 nm for all metals. The significant differences in number concentrations and morphologies are summarized in Table 1. Smaller particles are overrepresented in the micrographs of Fig. 2 due to the deposition via diffusion.

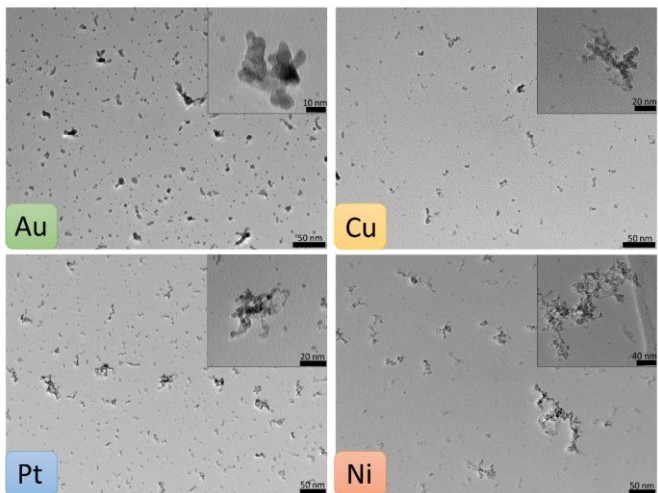

**Figure 2: TEM micrographs of spark generated Au, Cu, Pt and Ni NPs. The aerosol particles were collected via diffusion 1.3 s after generation for a collection time of 2h. Morphologies range between mostly spherical (Au) to increasingly fractal-like (Pt, Cu, Ni) particles.**

The particle morphology was assessed by calculating their circularity using equation 1, with values between 0 and 1. A circularity value of 1 corresponds to fully spherical particles and increasingly smaller values indicate more and more elongated shapes. Au particles are most spherical with an average circularity of about 0.9 (Table 1) whereas Cu and Ni have average circularity values around 0.55.

$$Circularity = 4\,\pi\,\frac{Projected\ Particle\ Area}{Perimeter^2} \tag{1}$$

Au formed fully spherical (i.e. coalesced) particles up to almost 6 nm whereas for Pt, Cu and Ni this threshold was at 3.2 nm, 1.3 nm and 1.1 nm, respectively. Fully coalesced particles were defined as primary particles by Tabrizi and coworkers (Tabrizi et al., 2009). The maximal size of primary particles was determined from particle circularities plotted against the projected area equivalent diameter for each metal (Fig. A1). The data was fitted with a logistic function and a decrease of the fit of 10 % with increasing size relative to the starting value was defined as the maximal primary particle diameter. 10% was 175 chosen to ensure persisting sphericity. For larger particles, aggregate (i.e. partially coalesced primary particles) and agglomerates (i.e. loosely combined particle assemblies) were observed abundantly. Thus, the sphericity is an effective parameter to quantify primary particle upper size limits.

**Table 1: Parameters to quantify particle morphologies for NPs generated via spark ablation collected after 1.3 s coagulation time and data on melting point depletion of NPs. Errors for circularity and fractal dimensions indicate standard deviations of all**

 analysed particles and error estimates of the maximum diameter of spherical particles (i.e., the projected area equivalent diameters) are detailed in appendix A.

| Metal | Average Circularity | Max. projected area equivalent diameter of spherical (*primary*) particles [nm] | Fractal Dimension $D_f$ of agglomerates (> primary particles) | Melting point for NPs (particle size 2 nm and 10 nm) and bulk [K] (reference) |
|-------|--------------------|--------------------------------------------------------------------------------|---------------------------------------------------------------|-------------------------------------------------------------------------------|
| Au | 0.9 ± 0.1 | 5.6 ± 0.3 | 1.75 ± 0.05 | 600, 1200, 1338 (Schlexer et al., 2019) |
| Pt | 0.7 ± 0.2 | 3.2 ± 0.1 | 1.76 ± 0.05 | 1287, 1685, 2043 (Wang et al., 2019) |
| Cu | 0.6 ± 0.2 | 1.3 ± 0.4 | 1.74 ± 0.06 | 964, 1289, 1357 (Wu et al., 2021) |
| Ni | 0.5 ± 0.2 | 1.1 ± 0.1 | 1.67 ± 0.07 | 893, 1623, 1726 (Van Teijlingen et al., 2020) |

Reasons for particle morphologies to differ between the four metals are manifold. Particle growth is strongly material, size and temperature dependent. Initial coalescence occurs within the spark generator under rapid cooling of the supersaturated vapor. However, coalescence and aggregation can continue well below the bulk melting points of the individual metals (Lehtinen and Zachariah, 2002). Significantly lower melting points for particles in the low-nm size range compared to bulk values have been estimated (see Table 1) because of the increased surface to volume ratio associated with large internal stress.(Buffat and Borel, 1976; Castro et al., 1990) As we generate particles as small as 1 nm this melting point depression likely has a notable effect on the particle morphologies observed here, resulting in a more pronounced coalescence and therefore a relatively late onset of agglomeration. Temperatures in Table 1 were obtained from literature using a thermodynamic liquid nucleation and growth model or molecular dynamics simulations. According to these models, Au NPs experience the lowest melting points (600 K for 2 nm particles; Table 1), which could explain the large diameter of primary Au particles of up to about 5.6 nm, whereas Pt exhibits the highest melting points, which might explain the lower primary particle size observed here. It is further reported that for Au particles, liquid-like behaviour, i.e., having a high mobility surface layer, occurs down to room temperature. (Arcidiacono et al., 2004; Kofman et al., 1994) Multiple studies describe Au particles smaller than approximately 3 nm as being liquid like at room temperature.(Castro et al., 1990; Feng et al., 2016; Pfeiffer et al., 2015)

For Cu and Ni particles a substantial melting point depression is estimated as well, but their primary particles are smaller than for Au and Pt. These elements are more prone to surface oxidation than Au and Pt (Barr, 1978; Payne et al., 2009), which results in higher melting points than estimated for the pure metals, preventing further coalescence/aggregation. (Gao and Gu, 2015; Olszok et al., 2023; Weber and Friedlander, 1997) The present oxygen impurities prevent the movement of the grain boundaries by pinning them (i.e., the thermal energy needed to increase the typically single crystal PPs in size is higher and thus prevents further growth). (Seipenbusch et al., 2003) Even though we use high purity (99.99%) electrodes,

oxygen impurities in the carrier gas in the low ppm range used in our study lead to the formation of oxide layers for Cu and Ni. (Hallberg et al., 2018; Olszok et al., 2024)

Other publications presented similar findings (Grammatikopoulos et al., 2014; José-Yacamán et al., 2005) as shown here and Feng and coworkers (Feng et al., 2016) developed a model to describe the evolution of the primary particle size and concentration.

Further particle growth via agglomeration can be described quantitatively via the fractal dimension $D_f$ of the particles (Table 1) (Eggersdorfer and Pratsinis, 2012; Olszok et al., 2021). $D_f$ values between $1.67 \pm 0.07$ and $1.76 \pm 0.05$ were determined for the four metals from TEM micrographs using box counting analysis.(Pashminehazar et al., 2019) These values are in good agreement with the literature for diffusion limited cluster-cluster aggregation of $1.77 \pm 0.03$.(Eggersdorfer and Pratsinis, 2012) This matches the findings of a previous study (Olszok et al., 2021) stating that spark generated aerosol particles are formed via diffusion limited cluster-cluster aggregation. Agglomerate morphologies align with NP structures found in previous publications (Debecker et al., 2024; Tabrizi et al., 2010).

### 3.1.2 Particle losses within the spark generator

Particle losses inside the spark generator were estimated for all four metals by comparing the mass of metal NPs collected on a filter for 6h immediately after the SDG with the mass that was ablated from both electrodes during the collection (Table 2). The electrode mass loss is lowest for Cu with 17 μmol and goes up to 82.5 μmol for Ni. The particle mass collected on filters ranges between 5 μmol (Au and Cu) and 37 μmol (Ni). Particle losses within the spark generator are substantial and range between 54% (Ni) up to 91% (Au). The majority of the losses can be attributed to losses immediately after NP generation mainly due to diffusion in the turbulent flow regime of the spark chamber. Additionally, the high electric and magnetic fields present in the plasma of the spark result in charging of larger particles and thus electrostatic losses. (Meuller et al., 2012; Schmidt-Ott, 2019; Tabrizi et al., 2009) Furthermore, collection efficiencies on the filters are close to 100% (Hinds, 1999) and increase slightly as the loading on the filter increases. This leads to a slight underestimation of the particle mass.

The total amount of ablated material strongly depends, among other factors described further below, on the thermal properties of the metals. Lower thermal conductivities lead to a less effective cooling of the electrode material and hence more evaporation. (Tabrizi et al., 2009; Tritt, 2005) Furthermore, lower boiling points also lead to more evaporation. Hence, the highest ablated electrode mass of Ni could be explained by the low thermal conductivity combined with a low boiling point. This large ablated mass of Ni explains the large number of agglomerates >20 nm seen in Fig. 2 Au and Pt mass loss is similar because Au has a lower boiling point but a higher thermal conductivity than Pt. Cu has a similarly low boiling point as Au but exhibits significantly less mass loss. This highlights again the strong influence of the thermal conductivity (which is almost 30% higher for Cu compared to Au) on the mass production as is also shown in (Loizidis et al., 2024).

**Table 2: Mass of ablated electrode material for each metal with data on boiling and melting points as well as thermal conductivities obtained from (Tritt, 2005). Mean and standard deviation of triplicate measurements are given.**

| Metal | Mass loss electrode [μmol] | Modelled mass loss electrode [μmol] | NP mass on filter [μmol] | Loss in Spark Generator [%] | Thermal conductivity [W m$^{-1}$ K$^{-1}$] | Bulk Boiling point [K] | Bulk Melting point [K] |
|---|---|---|---|---|---|---|---|
| Au | 54.5 ± 9 | 62.6 | 5 ± 0.5 | 91 ± 1 | 318 | 3073 | 1338 |
| Pt | 46.5 ± 2 | 42.1 | 9 ± 1.5 | 80 ± 2 | 72 | 4098 | 2043 |
| Cu | 17 ± 6.5 | 70.5 | 5 ± 0.5 | 66 ± 19 | 402 | 2848 | 1357 |
| Ni | 82.5 ± 8 | 55.7 | 37.5 ± 2.5 | 54 ± 4 | 93 | 3073 | 1726 |

The higher relative losses of Au, Pt and partially also Cu in the SDG could be explained by a lower vapor density in the electrode gap (i.e. less electrode mass loss compared to Ni) and thus slightly slower particle growth rates which lead to more diffusional losses. Ni vapor, however, is denser, particle growth is quicker and thus the diffusional losses smaller. There are
multiple studies investigating mass ablation rates (Domaschke et al., 2018; Schmidt-Ott, 2019; Tabrizi et al., 2009), most referring to the energy balance equation by (Jones, 1950). The equation was implemented for all four metals (Table 2) using a factor of 0.0005 as the fraction of the spark energy which is transferred to the hot spot (i.e. the location where the spark hits the electrode (Schmidt-Ott, 2019)); this factor is similar to what was determined by (Pfeiffer et al., 2014). For Au, Pt and Ni measured and modelled electrode mass losses agree well within a factor of 2 and for Cu the two values are within a factor of
about 4.

Furthermore, oxidation of the aerosol particles during generation as well as on the filter leads to an increase in total particle mass on the filter. This effect is most pronounced for the non-noble metals Cu and Ni but could also occur to some degree for Pt. Thus, the mass losses within the spark generator are likely underestimated.

**3.1.3 Aerosol particle size distributions**

Figure 3 shows particle size distributions determined from TEM micrographs for all four metals with modes ranging from 2 nm (Pt and Cu) to about 3 - 4 nm (Au and Ni) for the shortest coagulation time of 1.3 s and increasing modes for the two longer coagulation times. Particle sizes (i.e., the projected area equivalent diameters) are lognormally distributed as expected given the single-source of the particles.(Hinds, 1999) Total number concentrations for the shortest coagulation time are largest for Au and Pt and more than 60% lower for Cu and Ni.

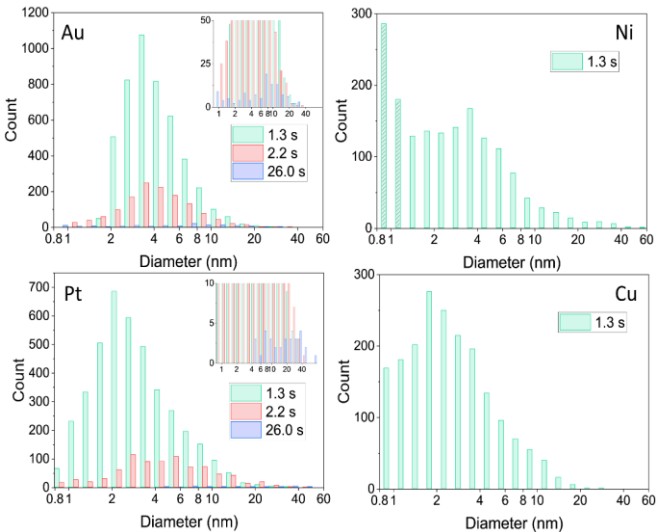

**Figure 3: Metal NP number size (i.e., the projected area equivalent diameters) distributions determined from TEM micrographs with ImageJ within one μm² of a TEM grid for three coagulation times: 1.3 s, 2.2 s and 26.0 s for Au and Pt and 1.3 s coagulation time for Ni and Cu. After a coagulation time of 1.3 s, particle modes range from 2 nm (Pt and Cu) to approximately 3 - 4 nm for Au and Ni. Particle concentrations are significantly higher (> 250%) for Au and Pt compared to Cu and Ni. The shaded bins for Ni indicate an overestimation of the smallest particles. Inserts for Au and Pt show the strongly decreased size distributions after a particle coagulation time of 26.0 s.**

For the longer coagulation times (2.2 s and 26.0 s) fewer particle numbers are observed due to increased coagulation (resulting in to a shift of the mode to larger diameters) and diffusional losses to the tube walls (appendix C). For Ni and Cu size distributions for the longer coagulation times could not be determined due to the overall smaller particle concentrations resulting in poor counting statistics. Difficulties in particle detection in TEM for Cu and Ni also arise due to the lower resolution of the NPs during imaging for these two elements, which depends, e.g., on the atomic number ($\propto Z^2$) and the

thickness of particles. Thus, given the lower atomic numbers of Cu and Ni (Z 29 and 28, respectively) and the smaller primary particles, Cu and Ni particles suffer from a lower resolution compared to Au and Pt (Z 79 and 78, respectively) particles. Although enhanced contrast can be achieved via image manipulation in ImageJ, the detection of particles with a diameter of a few nm is still challenging. The shaded bins in the case of Ni indicate the overestimation of the smallest particles due to the lower signal to noise ratio during particle detection with ImageJ.

Particles are deposited on TEM grids due to diffusion in a laminar flow regime. This results in an overestimation of the smaller particles compared to the larger particles and therefore the size distribution displayed in Fig. 3 is skewed towards smaller particle sizes compared to the size distribution present in the aerosol phase. This collection artefact can be corrected for, assuming that Brownian diffusion was the dominant particle collection process on the TEM grids. Thus, in determining the diffusion 'losses' (i.e. diffusion onto the grid) per size bin knowing the dimension of the collection chamber, the flow

rate and the diffusion coefficient of the particles, the particles number size distributions in the aerosol flow after a coagulation time of 1.3 s and 2.2 s, respectively could be calculated for Au and Pt (Fig. 4, black data). As expected, modes of the calculated size distributions in Fig. 4 are shifted by approximately 1-2 nm to larger sizes compared to the histograms in Fig. 3.

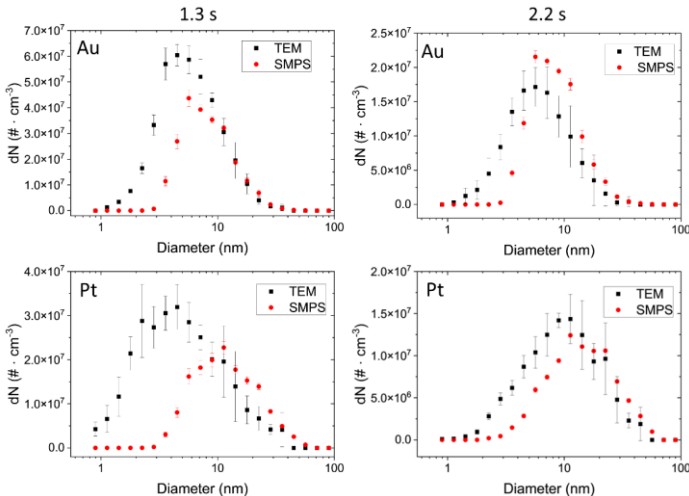

**Figure 4: Aerosol number size distributions estimated from TEM micrographs of Au and Pt NPs. The NPs were collected via diffusion for 2h for coagulation times of 1.3 s and 2.2 s and subsequently used to calculate the aerosol distributions (in black). In red, as a comparison, simultaneously measured SMPS distributions are plotted. Errors for the calculation consist of the propagate statistical error from triplicate measurements. SMPS error bars consist of the standard deviation of the scans taken for an 1h average.**

    Figure 4 also displays the aerosol particle size distribution measured with an SMPS (red data). For particle sizes > 10 nm the

concentrations calculated from TEM analysis and measured by SMPS align well for all four examples shown in Fig. 4. For particles smaller than 10 nm differences of the two size distributions become more pronounced the smaller the particle diameters are, due to the limited counting efficiency of the SMPS (DMA Model 3085, CPC Model 3776, TSI) at particle diameters below 5 nm. This leads to an underestimation of particles in this size range using the SMPS data. The TEM derived size distribution of Au and Pt for a coagulation time of 1.3 s, indicate that nearly 50% and 60%, respectively, of all

particles have a diameter < 5 nm. A comparison of coagulation times 1.3 s and 2.2 s shows the growth of the mode of the size distribution for Au and Pt particles as is expected for longer coagulation times due to Brownian diffusion. Hence, for a coagulation time of 2.2 s, the SMPS distribution aligns more closely with the calculated TEM distribution (Fig. 4).

Although TEM-derived particle sizes and SMPS measurements agree well for >10 nm particles, larger particles are affected by sampling uncertainties and thus lager errors in Fig. 4: Due to the smaller number concentrations of particles >10 nm counting uncertainties increase for these sizes. Where several hundred or thousands of <10 nm particles diffuse onto the grid per $\mu m^2$ within the sampling time of 2h, larger particles are only collected at significantly lower numbers on the grid (Fig. 3). Moreover, smaller particles can more easily be assumed to be spherical whereas particles >10 nm are mostly agglomerates. This results in an underestimation of the larger particles due to lower diffusion constants of non-spherical particles compared to spherical ones.(Moskal and Payatakes, 2006; Wang et al., 2017) Furthermore, uncertainties in particle diameters increase for larger, fractal-like particles because the assumption of circular shapes becomes less accurate..

For Ni and Cu, challenges in particle detection with ImageJ (see discussion above) prohibited a detailed analysis. However, the results of the analysis for coagulation time 1.3 s can be found in Fig. A3 in appendix A.

## 3.2 Metal nanoparticle coating on TiO$_2$ nanoparticles

### 3.2.1 Morphology of metal NP coating

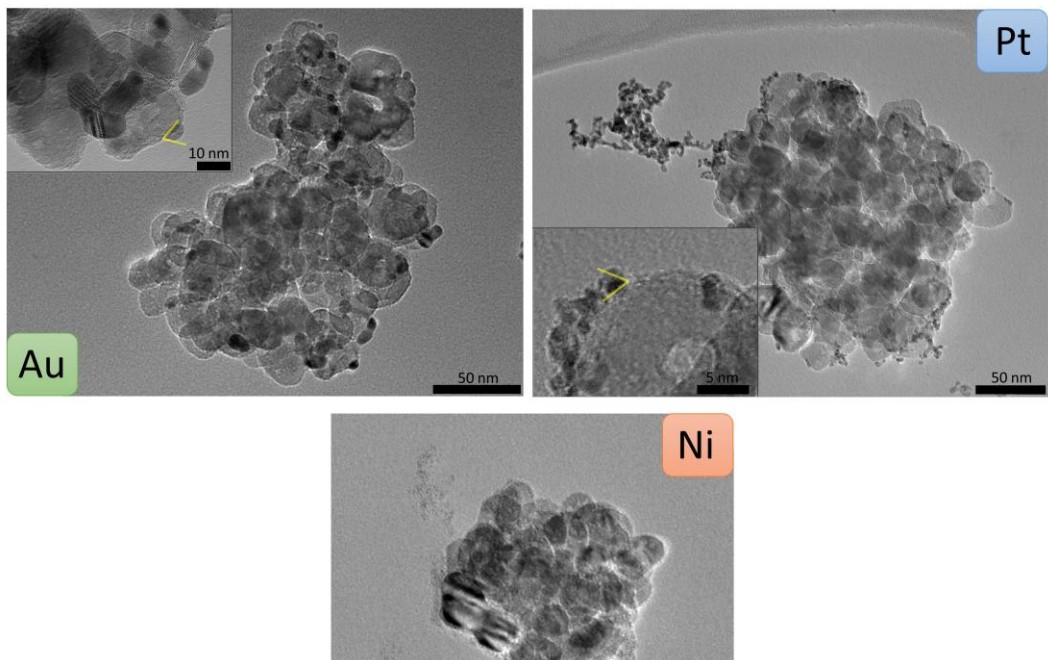

**Figure 5: TEM micrographs of TiO₂ substrate particles coated with Au, Pt or Ni NPs. Au and Pt NPs correspond to the small darker structures mainly visible at the edges of the substrate. Ni NP coating appears brighter than the substrate and is only visible on the left edge of the substrate particle. Inserts for Au and Pt provide details of individual primary particles and small agglomerates as well as indicated contact angles.**

All four metal NPs, generated as described above, were mixed and coagulated with nebulised TiO₂ substrate particles (mode 120 nm, concentration about $8*10^6$ particles $cm^{-3}$) for 26 s and analysed with TEM, STEM and EDX. TEM micrographs (Fig. 5) show significant differences in the coating with Au, Pt and Ni metal particles. Au and Pt exhibit a high density of 'island-like' coating, i.e. the presence of individual metal NPs on the TiO₂ surface, which are visible clearly at the edges of the TiO₂ substrate particle. Further towards the centre of the TiO₂ particles, the metal NPs are less visible due to the thick layer of TiO₂ particle which leads to a decrease in contrast for the metal particles.

For Au, the coating consists predominantly of primary particles and for Pt, a mix is observed between primary particles and agglomerates with sizes up to well above 50 nm. For Ni and Cu, no clear qualitative assessment of the coating was possible because these two metals produce significantly smaller primary particles in the spark generator (Table 1) and exhibit lower contrasts in TEM. While for Ni fractal-like particles are faintly visible at the edge of the TiO₂ substrate particle, no coating was detectable for Cu (Fig. B1). Thus, all further results and discussion on Cu can be found in appendix B.

The coating behaviour depends on various factors such as the thermodynamic properties of the two involved particle types and the size and number concentration of the coating particles. Surface wetting, i.e. the tendency of a liquid, or liquid-like particle to spread and adhere to a substrate particle (TiO₂), is a key concept to describe the coating. (Huhtamäki et al., 2018) Wetting depends on the particle size and the surface free energies of the involved species (Huhtamäki et al., 2018), i.e.

metals and $TiO_2$ in our study. Metal particles of a few nm in size can be considered liquid or liquid-like as discussed previously. When such particles coagulate with $TiO_2$, they can rearrange depending on the relative surface free energies of metal and $TiO_2$. If the metal has a lower surface free energy than the substrate species, surface wetting occurs, which can lead to the formation of a smooth coating layer. (Pfeiffer et al., 2015) Values for the surface free energies vary strongly in literature and depend on the techniques used to determine them, the crystal structures and the temperature and pressure of the measurement. $TiO_2$ has a surface free energy ranging between 0.6 and 1.3 $Jm^{-2}$.(Labat et al., 2008) Values for Au, Pt, Cu and Ni range between 1.4 $Jm^{-2}$ and 1.9 $Jm^{-2}$.(Kinloch, 1987; Tyson and Miller, 1977) Both, Au and Pt primary particles experience a slight wetting on the $TiO_2$ surface as can be seen in the inserts of Fig. 5. A contact angle (Young angle) of $67 \pm 4$ degrees for Au and of $60 \pm 4$ degrees for Pt was measured, i.e. well below 90 degrees, which was expected given that $TiO_2$ has a similar surface free energy as the metal NPs and therefore partial wetting rather than a smooth coverage is expected. The contact angle measurements are depicted in greater detail in appendix A (Fig. A2). Ni and Cu likely exhibit the same behaviour due to the similar surface free energies, however, this could not be confirmed from our TEM analyses as discussed above.

### 3.2.2 Size and concentration of metal NP coating

STEM micrographs and EDX spectra were recorded for Au, Pt and Ni NPs on $TiO_2$ particles to assess the number size distributions of the metal NPs that coagulated with $TiO_2$ (Fig. 6). EDX provides spectral maps which show the spatial distribution of the elements present in the sample and thus allows to distinguish between $TiO_2$ substrate and metal coating also in the centre of a $TiO_2$ particle. The EDX signal was integrated over 1h in order to achieve the highest possible resolution without disintegrating the particle due to exposure to the high energy electron beam. Figure 6 depicts the EDX recordings of the coating for Au, Pt and Ni (green) on $TiO_2$ particles (blue). In STEM micrographs (Fig. 6) the spatial distribution of metal NPs on $TiO_2$ particles is also clearly visible, particularly for Au and Pt due to the dependence of the contrast in STEM analyses on the atomic number of the analysed elements.

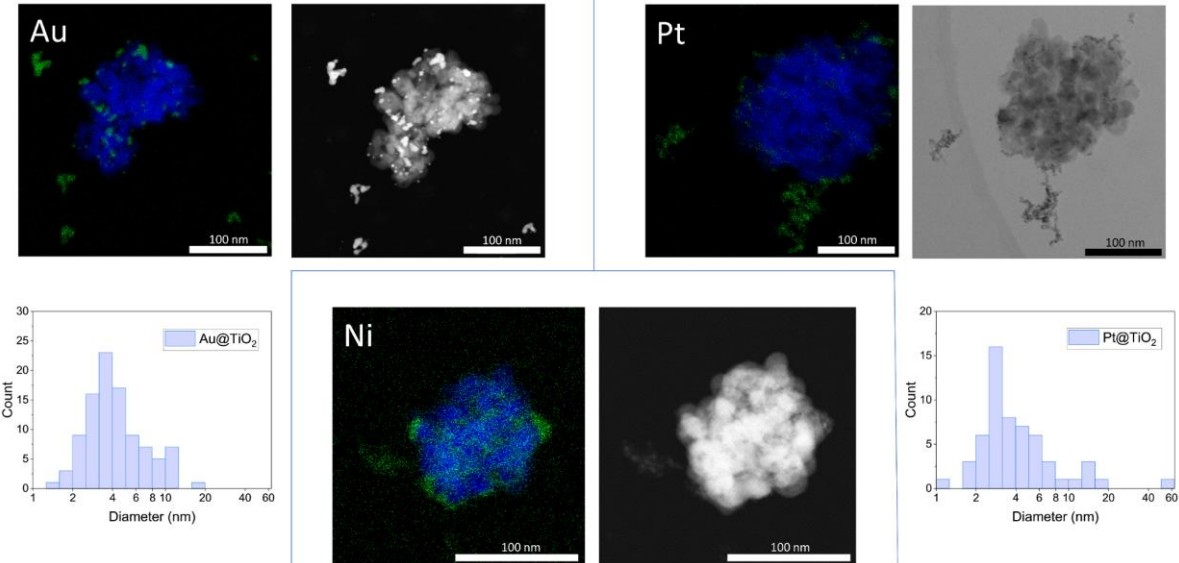

**Figure 6: STEM-EDX spectral mappings (TiO₂: blue, coating metals: green) and STEM micrographs (black and white) of TiO₂ substrate particles coated with Au, Pt or Ni. For Au and Pt, additional histograms of the number size distributions of the coating particles (i.e. 'Metal@TiO₂') are displayed.**

Au and Pt coating particles are lognormally distributed on the substrate with a mode of approximately 3.5 nm for both metals (see histograms in Fig. 6). This is slightly lower than the mode of the aerosol particle distributions of around 5 nm (Au) and

4 nm (Pt) determined in Fig. 4. These discrepancies can be explained by the particle size dependence of the Brownian coagulation where the large TiO₂ particles scavenge smaller metal particles with a higher efficiency than larger metal particles similar to the particle distributions on the TEM grid (Figs. 2 and 3) which overrepresent concentrations of smaller NPs.

The Ni NPs coating on TiO₂ could only be characterised qualitatively with EDX. Figure 6 shows >10 nm Ni agglomerates

attached to the TiO₂ surface, similar to the coating of Pt. Ni particles <10 nm are likely also present, but visualisation proved to be challenging given the smaller size of the primary particles compared to Au and Pt (Table 1).

The particle size distribution of metal NPs coated on TiO$_2$ was also assessed via TEM image analysis using the same

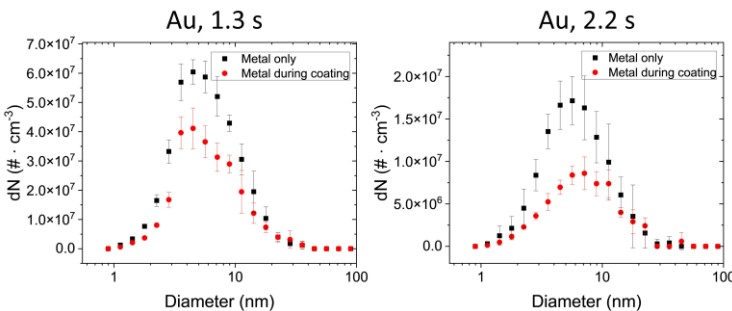

**Figure 7: Aerosol number size distributions estimated from TEM micrographs of uncoated Au NPs for mono-modal coagulation or bi-modal coagulation with TiO$_2$ NPs. For both coagulation times of 1.3 s and 2.2 s, the metal distributions during the coating (red, 'Metal during coating') were significantly lower than during experiments when TiO$_2$ was absent (black, 'Metal only'). This decrease in particle concentration is directly related to the TiO$_2$ particle coating with Au NPs. Error bars are shown for the propagated statistical error from triplicate measurements.**

approach as discussed for Fig. 4: The Au particle size distribution was calculated by counting the number and size of Au NPs on the TEM grids that did not coagulate with TiO$_2$ particles when Au and TiO$_2$ were mixed for 1.3 s and 2.2 s. This was

compared to the size distributions when no TiO$_2$ particles were present in the set up (Fig. 4). The Au number concentration during coating (red datapoints in Fig. 7) is significantly lower than in the case when no TiO$_2$ particles are present (black datapoints, Fig. 7). A reduction of the individual Au NPs (i.e. not attached to TiO$_2$ particles) of nearly 50% was determined ($4.0 * 10^8$ particles cm$^{-3}$ compared to $2.6 * 10^8$ particles cm$^{-3}$ for 1.3 s coagulation time). This reduced concentration of individual Au NPs can be attributed to their coagulation (i.e. coating) with TiO$_2$ and serves as a lower boundary for the

coating efficiency of $> 1 * 10^8$ particles cm$^{-3}$, i.e. about 20 Au NPs per TiO$_2$ substrate particle assuming $8 * 10^6$ TiO$_2$ particles cm$^{-3}$.

This is only a conservative estimate of the coating, because in the absence of TiO$_2$ particles (black data, Fig. 7) Au-Au NP coagulation is more effective, lowering the total particle concentration, compared to conditions when TiO$_2$ particles are present, where Au NPs coagulation with TiO$_2$ is a competing process to the Au-Au coagulation. Therefore, higher number

concentrations of primary Au NPs are coagulating with TiO$_2$ than estimated from the difference of the two number size distributions shown in Fig. 7. This underestimation of Au NPs coagulating with TiO$_2$ is confirmed qualitatively by the larger (i.e. >20) number of Au NPs counted on TiO$_2$ particles with STEM/EDX analysis (Fig. 6, histogram Au@TiO$_2$), where about 80-100 Au NPs per TiO$_2$ particle were counted. Equivalent analyses were conducted for Pt, Ni and Cu. Due to increased uncertainties in particle detection and conversion to aerosol distributions, no clear decrease in metal NP number

size distributions during coating can be determined for these three metals (Fig. A4).

To explore if a denser coating of metal NPs on TiO$_2$ particles could be achieved, we deposited NPs on TiO$_2$ films via diffusion. The films were exposed for 6h (Au and Pt) or 12h (Ni) to a metal NP flow. Lamellas, i.e., thin cross sections of

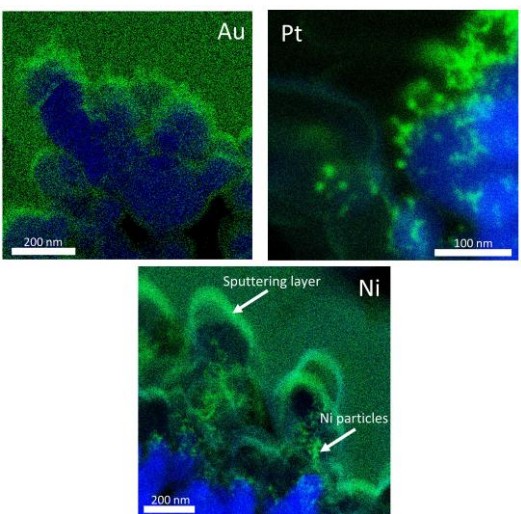

**Figure 8: EDX spectral mappings of the cross-section of TiO$_2$ substrate films (in blue) coated with Au, Pt or Ni NPs (in green). The TiO$_2$ films were exposed for 6h (Au and Pt) or for 12h (Ni) to a metal NP stream. White arrows in the Ni coating cross-section were added to distinguish between Ni NPs and the sputtering layer (see experimental section).**

metal coated TiO$_2$ films, were then cut. The longer collection time for Ni was needed to improve the signal intensity in EDX due to smaller primary particles and lower aerosol particle concentration of Ni compared to Au and Pt. Cross-sections of the

coated TiO$_2$ substrate layers were measured with STEM and EDX as depicted in Fig. 8. For Pt and Ni individual particles are visible (i.e. island-like coating, green in Fig. 8) which is in agreement with the observation of individual TiO$_2$ coating particles shown in Figs. 5 and 6. Au particles appear to aggregate the most, and thus a continuous layer forms on the TiO$_2$ surface (blue in Fig. 8) when exposed for 6h. As discussed above, this can be explained by the absence of an oxide layer which allows for partial coalescence and necking even at room temperature and thus facilitates the formation of a continuous

layer. Pt and Ni show similar coating behaviours for particles >10 nm. In contrast to Au, Pt and Ni both exhibit strong agglomerate formation with particles up to several hundreds of nanometres in size and even at long exposure times of 6h (Pt) or 12h (Ni), no continuous metal layer forms on the TiO$_2$ particle layer.

The smooth green-coloured area in the EDX mapping of Ni in Fig. 8 is due to the Au sputtering layer on the TiO$_2$ film (see experimental section) and not due to Ni NPs. Only the faint green fractal structures in the lower half of Fig. 8 corresponds to

Ni NPs, loosely attached to blue TiO$_2$ particles (see arrows in Fig. 8). In addition to the EDX recordings presented here, STEM micrographs can be found in Fig. A5.

**4 Conclusion**

In this study we investigated spark discharge generated metal NPs and their coating behaviour on $TiO_2$ substrate NPs in the aerosol phase. NPs of four metals (Au, Pt, Cu, Ni) were characterised for size and morphology. Using TEM measurements,
aerosol particle number size distributions of particles as small as 1 nm with modes of the size distribution of $3 - 5$ nm were determined, which poses a significant challenge for commonly used aerosol particle size measurement techniques. Differences in particle size and number distributions for the four metals could be correlated with their thermodynamic properties such as melting point combined with their oxidative properties.

Coating of the four metal NPs on $TiO_2$ substrate NPs via coagulation was characterised with TEM, STEM and EDX and
showed that Au and Pt particles attached to $TiO_2$ particles partially wetted the $TiO_2$ substrate. Up to about 100 Au NPs were coating a single $TiO_2$ particle. For Au, a continuous coating layer could be achieved by increasing the deposition time to several hours. A detailed characterisation of metal NPs and their coagulation and coating behaviour with $TiO_2$ particles as provided here will be important to assess chemical or electronic properties of coated $TiO_2$ particles in future studies.


**Appendix A**

Additional and complementary graphs supporting the paper are presented here.

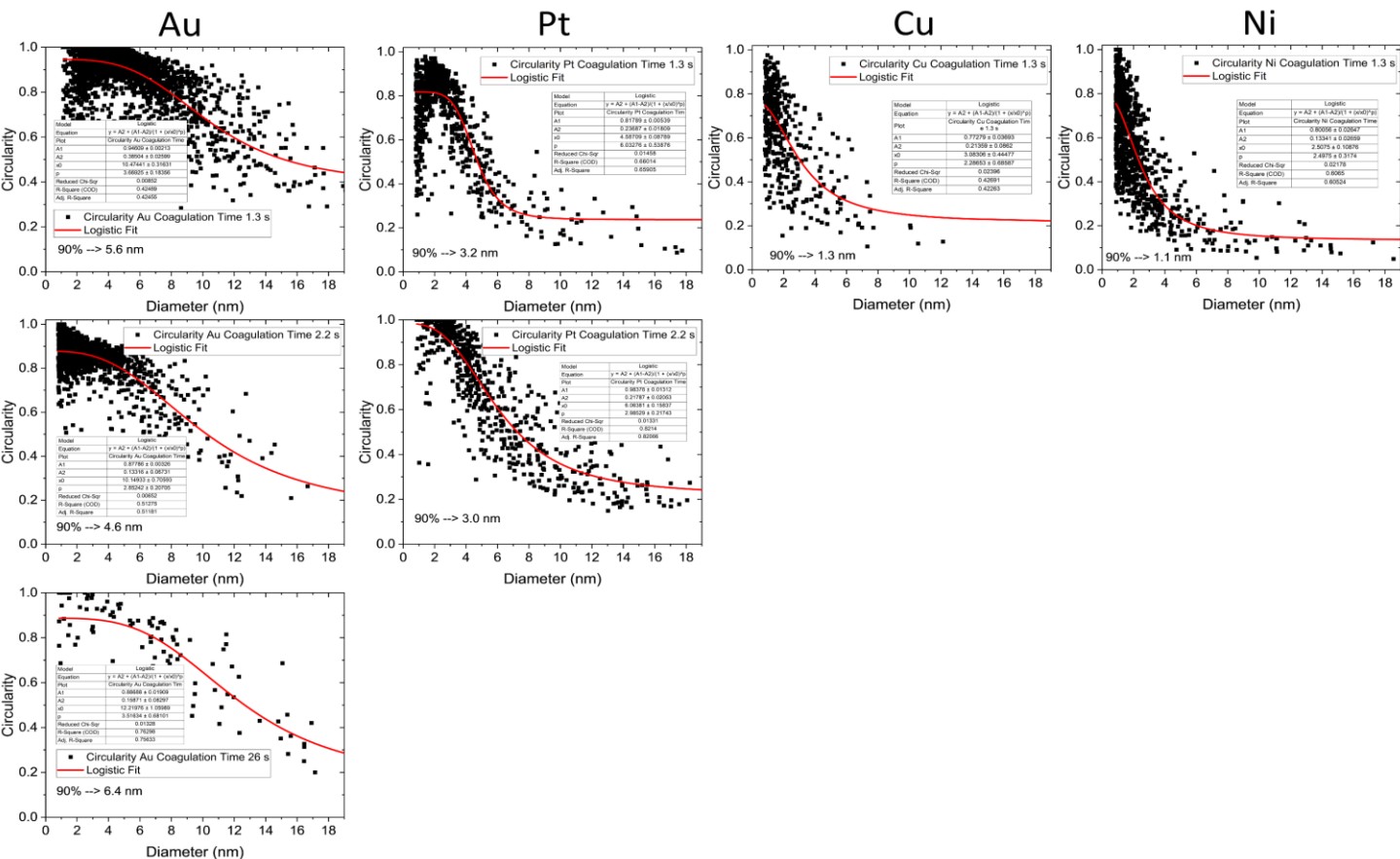

**Figure A1: Particle circularities plotted for the metals Au, Pt, Cu and Ni against the surface equivalent diameter after 1.3 s coagulation time and after 2.2 s and 26.0 s for experiments where particle numbers were sufficiently large. The data was fitted with a logistic function to determine a maximal primary particle size for each metal. While Au particles remain spherical up to a size of more than 5 nm, Pt, Cu and Ni begin to form agglomerates below 3.5 nm. Errors given in Table 1 are derived from the errors of the fit parameters shown in this figure.**


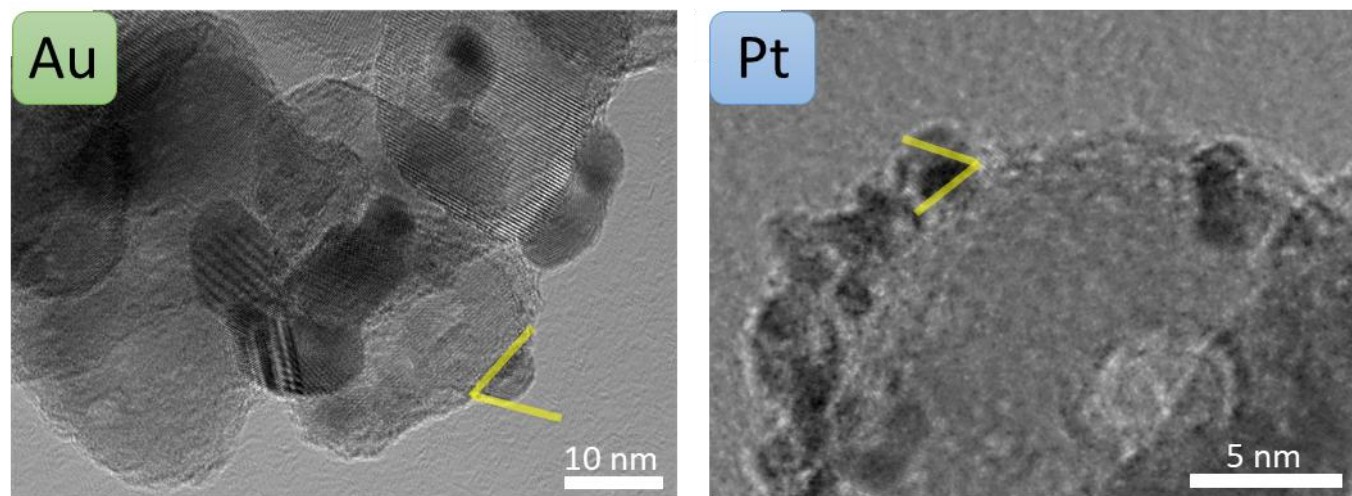

**Figure A2: Contact angle measurements for Au and Pt coated on TiO₂ complementary to Figure 5. Contact angles of 67 ± 4 degrees and 60 ± 4 degrees were determined for Au and Pt respectively.**

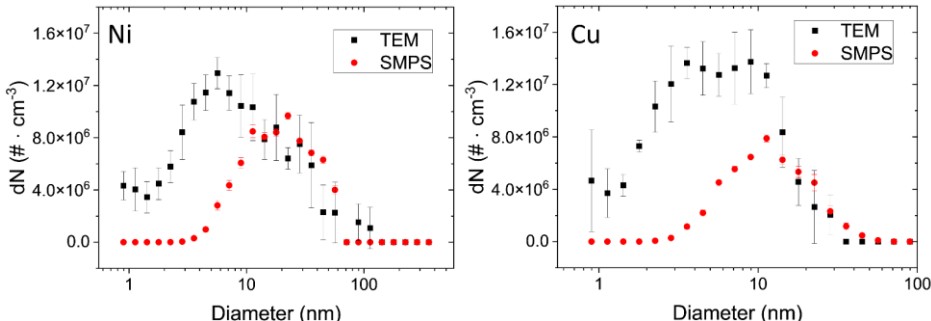

**Figure A3: Aerosol particle number size distributions calculated from TEM micrographs of Ni and Cu NPs. The NPs were collected via diffusion for 2h for a coagulation time of 1.3 s and the corresponding aerosol particle distributions were calculated as described for Fig. 4. In red, as a comparison, simultaneously measured SMPS particle size distributions are plotted. The calculated distribution matches the measured distribution well for particles > 10 nm. For < 10 nm particles, the calculated distributions deviate from the SMPS values which can be partially attributed to the limit of detection of the SMPS instrument.**

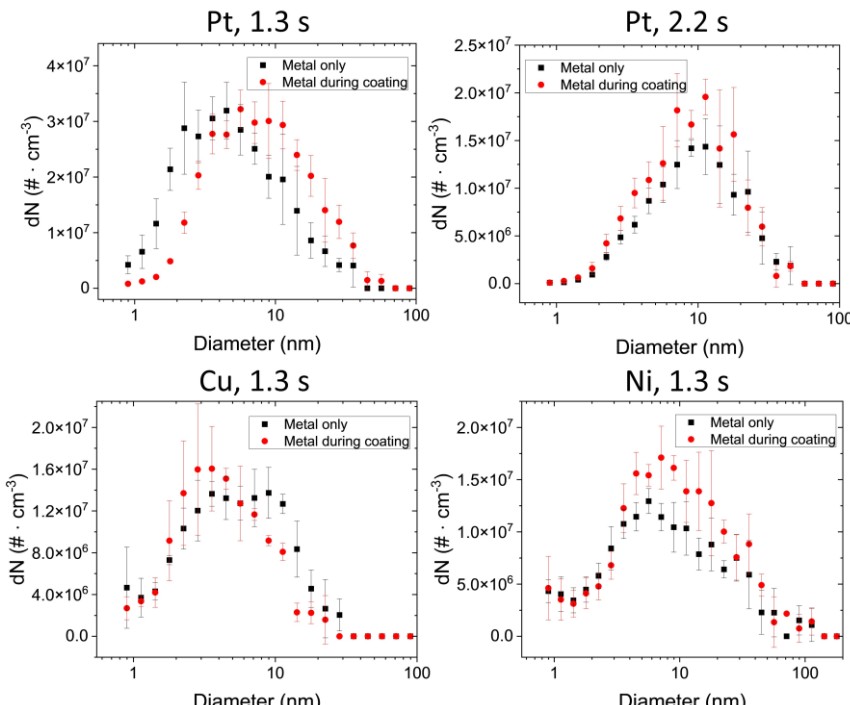

**Figure A4: Aerosol number size distributions estimated from TEM micrographs of uncoated Pt, Cu and Ni NPs for mono-modal (in black, 'Metal only') or bi-modal coagulation with TiO₂ NPs ('Metal during coating', in red). No clear difference in particle number concentration and size is visible between the two size distributions for the three metals. Errors for the calculation consist of the propagated statistical error from triplicate measurements.**

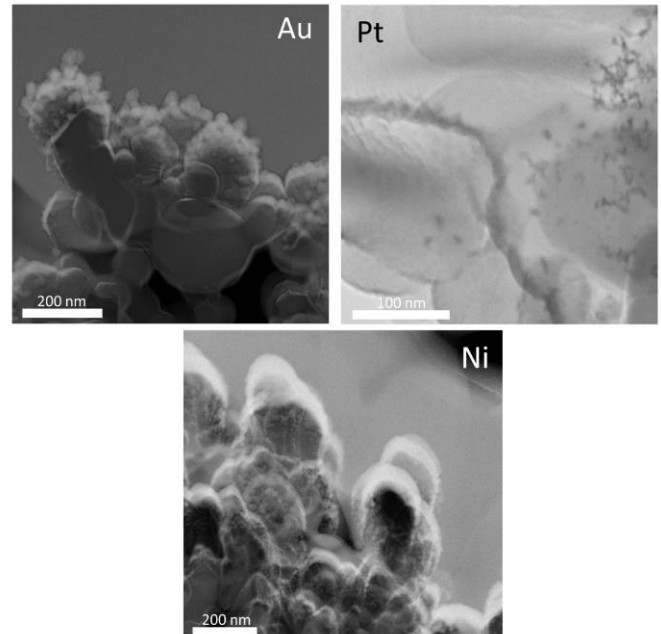

**Figure A5. STEM micrographs of lamellas of coated TiO₂ films for Au, Pt and Ni. While coating particles are clearly visible for Au (bright dots) and Pt (dark dots), Ni particles are only visible in EDX mappings (see Fig. 8, main text). The bright caps in the Ni micrograph originate from a strong background signal due to sputtering with Au. For details, see the method section of the paper.**

**Appendix B**

Given the low number concentrations and small sizes of Cu aerosol particles (see Fig. A3, Fig. 3 and Table 1), the detection of Cu coating on TiO$_2$ particles was not possible (see Fig. B1 below).

Likely, Cu particles present are below the limit of resolution for TEM, STEM, EDX and are thus not visible in Fig. B1 (A - C). The larger background signal of Cu in EDX and STEM (Fig. B1, D, E) is due to Cu-containing components inside the microscope, causing higher background levels compared to the other three metals. Therefore, it was not possible to analyse

Cu NP coatings. However, given the slightly lower surface free energy of Cu (Kinloch, 1987) compared to TiO$_2$ (Labat et al., 2008), we estimate island-like coatings of predominantly primary particles.

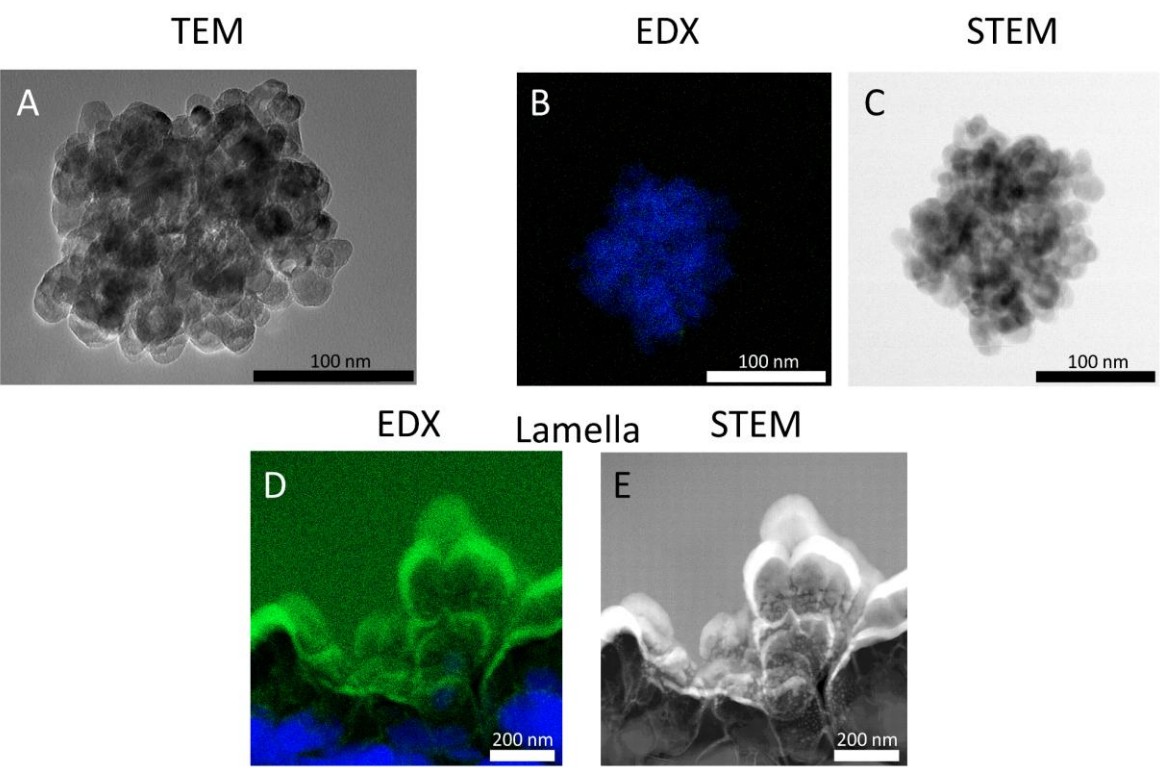

**Figure B1: Attempt of electron microscopic imaging of Cu NP coating on TiO$_2$ particles (A-C) as well as of the deposition of Cu NPs on a TiO$_2$ film for 12h and subsequent lamella extraction (D, E). While in TEM (A) no coating is visible, a very weak signal of Cu in EDX (B, in green) likely indicates that coating particles are present on the surface of the TiO$_2$ particle. However, Cu NPs cannot be resolved in STEM (C). EDX of the lamella (D) shows a strong Cu signal which originates mainly from Cu components in the instrument and not from Cu NPs. Thus the coating behaviour of Cu on TiO$_2$ particles can not be assessed reliably with the techniques used here.**

**Appendix C**

Complementary to the experimental approaches detailed above, a numerical coagulation model from Zhang and coworkers (Zhang et al., 2020) was adapted and implemented to determine the decrease of Au NPs due to coagulation with $TiO_2$ particles and diffusion to tubing walls, respectively.

Input for the model was the aerosol size distribution at the point where the Au NPs are mixed with the $TiO_2$ substrate particles ($8*10^6$ $cm^{-3}$, mode at 120 nm). The Au NP input distribution was determined via TEM analyses and a subsequent calculation of the corresponding aerosol concentration (see main text, Fig. 4). The $TiO_2$ distribution was measured with an SMPS.

Figure C1.A shows how many particles per size bin are lost due to coagulation during the coating. For this calculation, the Au NP concentration before coating (t=0 s) is compared to the modelled concentration after a specific coagulation time. For 26.0 s up to approximately 20 nm, nearly all of the particles coagulated. Less pronounced losses are observed for the much shorter coagulation times of 1.3 and 2.2 s, as expected. Although the modal model does not include diffusion losses which are highly relevant for < 50 nm particles, they are accounted for separately via a particle loss calculator (von der Weiden et al., 2009) (Fig. C1.B). The diffusive losses were determined for all three coagulation times. Results from these calculations (Fig. C1.B) show the importance of diffusive losses, especially for <10 nm particles. For an accurate assessment of the particle losses in our experiments, a combined coagulation and diffusional loss model would be needed, which will be presented in a forthcoming study.

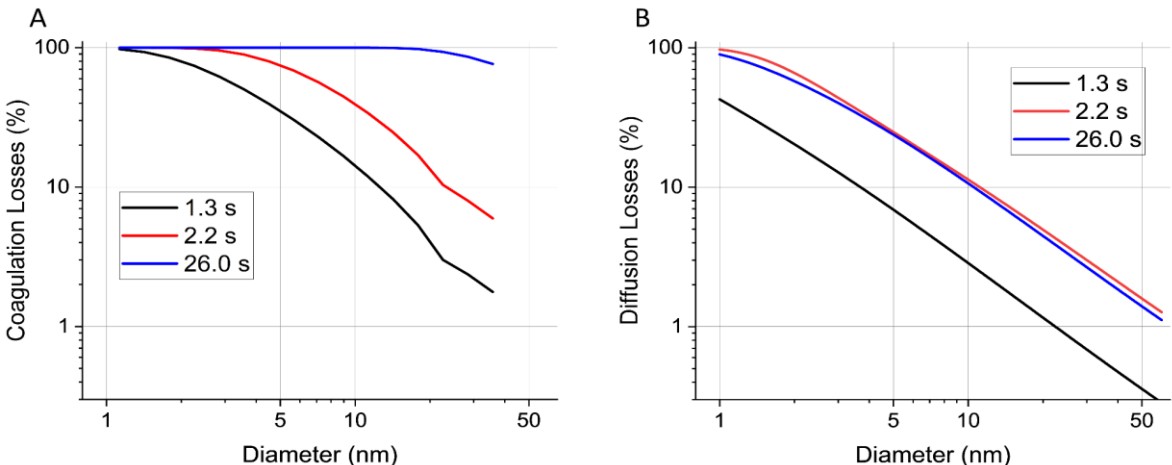

**Figure C1: A: Modelled 'losses' (decrease in particle concentration) due to coagulation during coating of Au on TiO₂ after coagulation times 1.3 s, 2.2 s and 26.0 s. B: Size dependent diffusion losses for the three coagulation times. Losses for 1 nm particles are about 10 times higher than for 10 nm and about 100 times higher than for 50 nm particles for all coagulation times. 2.2 s (long, small diameter tube) and 26.0 s (shorter, larger diameter tube) have similar losses given the non-linear (i.e. exponential) dependence of the diffusion losses on the tube length.**

**Data availability**

Data for this article, consisting of Excel files are available at the Open Science Framework at https://doi.org/10.17605/OSF.IO/ZM3G4.

**Author contributions**

Conceptualization – MB and BG; investigation – BG, MB and AA; methodology (adaptation and model implementations) – NB and BG; methodology (image analysis) – BG; methodology (electron microscopy) – MW, BG and MB; project administration – MB and BG; funding acquisition and supervision – MK; writing (original draft) – BG; writing (review and editing) – all authors

**Competing interests**

The authors declare that they have no conflict of interest.

**Acknowledgements**

This work was supported by the Swiss National Science Foundation (200021_192192/1).

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
