# Peer review of "Spark ablation metal nanoparticles and coating on $TiO_2$ in the aerosol phase"

_Aerosol Research, 2025_

## Referee Comment (RC2)

Overall the work is written in a vague manner without direct means for relevant confirmations. This work also made the impression for reproducing already published works published years ago but not yet fully cited them. This "gave the chance" of this work to pretend to be the initial discoveries. The morphologies of the NPs demonstrate that this study did not yet optimize the system. Only switching on the generator and collecting them for subsequent analysis cannot add scientific values, particularly ignoring the large volume of literature, which reported almost the same line of argumentations. The authors also try to use artificially texted letters to mark the different metals without showing any experimental proofs. Raising such concern is mainly due to all their similar appearances but undistinguishable nature from the TEM results.

Large body of relevant literature was severely missing.

Primary particle was defined scientifically incorrect.

where the authors showed the 1-nm particles?

Melting point of the 4 metals cannot be directly used to evaluate their different particle sizes, as the impure surface of the particles also makes dramatic influences.

How the authors prove that the oxidation took place during NP production?

Why the mass was evacuated using micro molar? As the authors also claimed that the SDG can also deliver NPs of high quantity. How the amount of micro molar supports the aforementioned claim?

For the different production rates of the metals, previous study already clearly explained why. Please cite the relevant literature, not only saving the unnecessarily repeated work but also not making an impression of confined literature review.

The authors also argued that the mismatch of size distributions measured between the TEM analysis and SMPS data is mainly due to the low counting efficiency from the latter. How to explain the overestimation of diffusional deposition of smaller NPs for TEM analysis?

A very puzzled presentation for Fig. 5 onwards is to remove the Cu NPs. Could the authors provide any scientific reasons for such inconsistencies?

How the edges of the NPs were determined when using eq. 1 to calculate circularity?

How the authors separate the ones coated over $TiO_2$ with those of self-coagulation?

Other mistakes also showed the carelessness and this can also give the reviewer for scientific suspect. Wrong spelling in page1, "Department od Environmental Sciences" should be "Department of Environmental Sciences".

---

## Author Comment (AC1)

Answers to comments of reviewer 1

The manuscript (MS) describes the production and characterization of aerosol nanoparticles (NP) as single material form and as hetero-aggregates (HA) using a spark discharge generator (SDG) combined with a nebulizer. The HA consist of TiO2-NP, on which four different metal NPs are deposited. The pure metallic particles and the HA are characterized online and offline using a variety of methods such as SMPS and TEM-EDX.

The authors like to thank the reviewer for their constructive comments. Our answers to their questions and comments are given below in blue font.

**General comments**

The MS contains many valuable and original results, which, however, still need to be bundled and formulated much better. In particular, there is no common thread linking the research question addressed, its answer and further consequences of the results. What are, up to the conclusion, new findings the authors want to report about the coating of larger NPs with metal NPs from spark ablation? The authors are encouraged to thoroughly rewrite the manuscript with a clear story line and targeted motivation. One interesting part to focus on could be the metal-depending layer quality (layer porosity, thickness, contact angles etc.) of spark generated NPs on TiO2 surfaces. In this context, the oxide affinity of certain metals, e.g. Au vs. Cu or Ni, could be systematically investigated, including many of the already presented data in a condensed way. The individual points are explained in more detail below, with the page (P) and line (L) indicated.

In the revised manuscript we describe in more detail the motivation of our study and the open questions our results address (line 73-92), see also comment below, point 1.
We like to point out that we present here a study with a strong focus on the methodology of metal nanoparticle production and the characterization of their number size distributions. Therefore, the related research aims are also largely method related.

1) Introduction
   The spark ablation section is well described; however, a detailed motivation for coating TiO2 NPs with spark generated particles is missing. Topical literature regarding this aspect is missing, too. While the introduction covers many interesting topics a clear focus of the presented work at the end of the introduction is missing. i.e. which open research questions should be addressed here. This goes beyond the listing of the performed investigations and covers rather the coherence of the results presented here.
   We now added a more detailed paragraph at the end of the introduction detailing the motivation of the presented study and the open research questions they address. (line 73-92)

2) P 4, L 104-108: Collecting small particles by diffusional deposition leads to a size-biased representation of the particle, which is specially agravated for small nanoparticles as outlined in the MS. However, also the amount of deposited particles differs substantially from the number of particles in the aerosol. Therefore, the question needs to be discussed how the fraction of nanoparticles sampled on the TEM grid (and on the Teflon filters) in relation to the total aerosol particles is determined. It is surprising that the number of deposited particles and aerosol particles are so close to each other (e.g. Fig. 4). Which

approach was taken to quantify the fraction of deposited particles? Please explain this point in more detail.

We calculate the number size distribution of the particles in the aerosol phase by combining the measured number size distribution derived via counting of the TEM pictures and the calculated diffusion of particles from the aerosol flow onto the TEM collection grid. To calculate this particle size distribution in the aerosol phase the following parameters were required: The dimension of the collection chamber, the diffusion coefficient of the particles and the flowrate of the aerosol stream. The detailed analysis can be found in the available excel files on the data repository. First, the 'diffusional losses' onto the TEM grid are calculated following the method described by von der Weiden et al. (2009), i.e. calculation of losses in a tube with the flowrate and the length of the TEM grid. After calculating the fraction of particles deposited from the aerosol phase onto the grid, the total amount of particles passing over the grid can be calculated. This is now stated in revised version in more detail in line 287-290.

3) How does a quantitative NP sampling work on Teflon filters with 2 µm pore size and NPs being smaller than approx. 50 nm?

The primary collection process is diffusion, thus, sampling efficiencies of nearly 100% are reached for particles below 100 nm. (see e.g., W.C. Hinds, Aerosol Technology, Wiley, Second Edition, p. 202 – 203, Figure 9.12; second edition). This reference is now explicitly mentioned in line 241-242.

4) P 5, L 126: What is the motivation for depositing agglomerate NPs on a solid TiO2 film and the subsequent sputter coating with a 100 nm Au layer? It is not clear why such a thick platinum layer needs to be applied. Does this not lead to a significant loss on resolution?

We changed the description of the lamella experiments (line 148-157).

5) P 6, L 151ff: The idea of employing the circularity to define the spherical particles is very good. This approach could make an important contribution to the issue about the first nucleating clusters and their further fate. This approach is certainly a highlight of the MS and would deserve some more evaluation. Why was the limit of 90% also applied for sphericities staring well below 1.0 (cf. Fig. A1, e.g. Pt for 1.3 s residence time)? Is the fact that some particles start already with sphericities well below 1.0 an indication that in fact they consist of much smaller units, i.e. atomic clusters? In Fig. 3 for Pt 3 residence times were analyzed with respect to size distribution. Why was the results for 26 s not also included in Fig. A1 in the Pt column?

Why was the limit of 90% also applied for sphericities staring well below 1.0?

Primary particle (PP) size differs greatly for the different metals (see Fig. 2). The smaller the maximal PP size is, the more pixelated the analyzed PPs are in image processing. This effect, which is most pronounced for Cu and Ni but was also observed for Pt can lead to an underestimation of the circularities (i.e. a pixelated, elongated shape rather than a smooth spherical structure; for the present analyses 2.8 pixels per nm resolution were determined). Thus, the sphericity of the detected PPs can be lower than one. As the pixilation is size dependent (i.e. more pronounced for smaller particles) and thus also metal specific, no absolute circularity value can be defined for which on average full sphericity occurs. In the case of Pt 1.3 s this is clearly visible by the broad circularity distribution below 2 nm.

Due to these uncertainties, we introduced an estimated 10% limit of the drop of the fitted

line as limit value for particle sphericity.

Why does for Cu and Ni sphericity start below 1.0, i.e. due to atomic clusters?
The observation that the sphericity for Ni and Cu starts below 1.0 is mostly due to analysis limitations discussed above, i.e. pixel size.

Why were the results for 26 s not also included in Fig. A1 in the Pt column?
No spherical primary particles were found anymore on the grids for a 26 s coagulation time, but only large agglomerates. Therefore, no curve could be fitted and no PP size could be determined. For this reason, the plot is not shown in Fig. A1.
This is mentioned in the caption to figure A1.

Additionally, one sentence is added in line 192-193 highlighting the suitability of using the sphericity as a parameter to quantify primary particle size limits.

6) The authors state that the coagulation time influenced the primary particle size of the respective materials. How is that possible when primary particle formation is completed approx. 100 µs after the spark discharge? And how can the sphericity of the Pt increase with longer residence time (from 1.3 s to 2.2 s) in Fig. A1?
Given the discussion in 5) above, the data does not definitively indicate that particle sphericity increases with increasing coagulation time, which is due to the pixilation and resulting uncertainties as mentioned in 5).

7) P 7, L 155 and Table I: The authors refer to "primary particle size" and "agglomerate particle size". Please include a detailed description which size (primary vs. agglomerate) is shown in the size distributions. What equivalent diameter is presented here?
All diameters presented are projected area equivalent diameters. They are defined in line 144 of the manuscript. This definition is now also added in Fig. 1, Table 1 and also further below in subsection 3.1.3 (Line 268)

8) P 7, L 169-171: The authors state that Au NPs might form oxides as a consequence of using N2 5.0 with oxygen impurities. This argument holds certainly for Cu and Ni. Pt NPs from spark ablation can exhibit thin oxide layers on the particle surface; however, for Au, an oxidation is impossible under the mentioned experimental circumstances. This fact supports, in turn, the relatively large primary particle size of <6 nm for Au NPs with a circularity of C=1 that is mentioned by the authors.
We agree with this statement. We deleted our comment that of Au particles could form oxides. (L 264)

9) Besides the low melting point, especially the absence of oxidation of the surface of Au NP contributes to the strong necking and coalescence growth of Au primary particles/clusters. On the other hand, oxygen-sensitive materials such as Ni and Cu experience a so called "pinning effect" by adsorption/reaction of the surface with oxygen (e.g. Seipenbusch et al. J. Aerosol Sci. 34, 2003). Such a pinning effect can increase the activation energy for coalescence from typically 50 kJ/mol (clean metal surface) to about 80 kJ/mol (partly oxidized metal surface, e.g. Ni). This point should be more elaborated here.
Thank you for the explanation, we were not aware that this effect. We added a sentence

*explaining this effect (line 217-219) in the revised manuscript and added the reference mentioned above.*

10) P 8, L 193-195: Regarding the generally poor charging of very small NP it is surprising that losses due to electric fields should accumulate to such an important amount. Are the substantial losses in the SDG chamber not rather related to the high particle diffusivity?
*We added this in the MS in line 237-242.*

11) P 8, L 206-208: The authors mention the ejection of micron sized particles during spark erosion and a subsequent deposition of those within the housing of the spark discharge generator. The observation of Tabrizi et al. is very interesting and is also found for other processes where metal surfaces are locally heated up by sparks or pulsed lasers. It is also reasonable to attribute a high mass loss to the deposition of large micron sized particles. However, since the significance of this mass loss channel is expected to depend very much on the material properties, a more direct confirmation of these droplet-based particles would improve the convincing power of the argument. For instance, samples from the SDG chamber could be taking a wipe sample and analyzing it with electron microscopy (SEM/EDX or TEM/EDX) for micronic spherical particles.
*This would indeed be an interesting way to check if micron size particles are present in the spark chamber. However, as this is not the main focus of the manuscript, such measurements were not performed.*

12) P 12, L 281: The authors derived contact angles for Au and Pt on TiO2. Please show a TEM image with a magnification where the contact angle is outlined. Please increase also the size of the TEM micrographs shown in Fig. 5. The inserts are very hard to see.
*The size of Fig. 5 was increased such that the contact angles are now clearly visibly. Furthermore, contact angles are drawn in the same figure and the corresponding angles are inserted. The micrographs with the drawn angels were also enlarged and included in the appendix (Fig. A2).*

13) P 13/14/18, Fig.6, 8, B1: The coloring of the EDX maps is confusing. Please use a defined color for each element to distinguish, e.g., background from sputter layers from NPs from TiO2 support. Fig.6: The authors present dark field (DF) and bright field (BF) TEM(STEM) images. Please refrain from using BF images for Pt@TiO2 since Pt has a high contrast in HAADF, such as Au in Fig.6 top left.
*In all EDX recordings, the metal is colored green and the $Ti(O_2)$ is colored blue consistently. Sputter layers are not visualized; the green color in the Figures B1 solely refers to Cu. The large features which are also colored green are also originating (falsely) from the Cu signal. This is, as discussed in the manuscript (e.g. in the caption to Fig. B1), due to a strong Cu signal caused by Cu parts within the instrument. We are not showing the sputtering layers as they are not of relevance.*

*We are aware that HAADF usually gives better contrast than BF for high Z elements. However, the analyses we obtained for Pt on $TiO_2$ using BF detection resulted in a better resolved STEM image than the respective HAADF image. Due to the thickness of the $TiO_2$ particles and the difference in density of Ti and Pt, it was difficult to correctly set the intensities for the recordings and thus, as the Pt particles are visible better with the BF, the BF image was chosen to be shown in the manuscript.*

14) P 15, L 328: The authors mention a continuous layer of Au on the TiO2 substrate after deposition. This observation can be related to the oxygen affinity of the metals used. Due to the absence of oxygen in Au NPs, partial sintering and necking can be observed even at room temperature.

We thank the author for this comment and added a sentence describing the reason why a continuous layer of Au forms on the $TiO_2$ substrate (due to the absence of oxygen in Au NPs, partial sintering and necking at RT) (line 387-389).

15) P 19, L 370-371: The calculation of the corresponding aerosol concentration based on the TEM micrograph is still not clear (cf. comment above). How was this done?

See 2). Details were added in the main text as described in 2).

16) P 19, Caption of Fig. C1: The diffusion losses in laminar flow in tubes does not depend on the tube diameter! Therefore, this argument about the influence of the larger tube does not apply. There must be another reason for the similar losses at different residence times since the amount of loss depends critically on the duration.

Thank you for pointing this out. It is correct, there is no dependence of the diffusion losses on the tube diameters. The reason why particle losses of the 2.2 s and 26.0 s are similar is due to how the losses accumulate non-linearly (in fact, exponentially):

In the beginning, when the concentration is high, way more diffusional losses occur compared to when the particles passed already a certain length through the tubing. So, the higher total length of the 2.2 s configuration does not result in much more losses than in the shorter 26.0 s configuration.

The caption to Fig. C1 was changed to highlight the above described.

**Literature**

von der Weiden, S.-L., Drewnick, F., and Borrmann, S.: Particle Loss Calculator – a new software tool for the assessment of the performance of aerosol inlet systems, Atmospheric Measurement Techniques, 2, 479–494, https://doi.org/10.5194/amt-2-479-2009, 2009.

---

## Author Comment (AC2)

The paper discusses the deposition by coagulation of small particles generated by spark discharge onto a TiO2 nanopowder to make an unspecified catalyst. Spark ablation is known to generate small particles, but a reliable method to synthesize useful catalysts with them is still lacking. The paper goes in more technical detail than a recent publication with a similar approach (Debecker 2024), and provides several useful insights, and novel approaches compared to the state of the art. The paper does not mention catalytic performance, but focuses on the aerosol synthesis route, and as such is a suitable topic for this journal.

The authors like to thank the reviewer for their constructive comments. Our answers to their questions and comments are given below in blue font.

**General comments**

1) The introduction seems written as an afterthought: It contains a lot of puzzle pieces, but the reader is left to guess what the motivations are for including these pieces. A clearer focus here helps highlight which parts of the work are novel, and which ones are not.
   The introduction was expanded and the main motivation of our study and open questions addressed are now clearly described (line 73-92). Furthermore, the abstract was also expanded to state these research questions clearly.

**Technical issues:**

2) Residence time for the different experimental parts aren't completely clear. E.g. Fig 1; L100: The residence time of volume C is specified. But what is the residence time before and after volume C? Figure 2 shows particle size 1.3s "after generation": depending on where you define generation, this can be somewhere before, in, or after volume C…
   We define the three residence times as defined by all the tubing and mixing volumes, through which the particles pass starting from the exit the spark generator to particle sampling point.
   The corresponding text in the manuscript was adapted as suggested by the reviewer (line 120-125).

3) The description of Fig 1. implies coated particles are not analysed (either/or?). Is the deposition tree for coating experiments the same as direct particle deposition without volume C? Were flows/volume adjusted to maintain correct residence time?
   The flow was adjusted in the case of the bi-modal coating. The caption to Figure 1 and the text in section 2.1 (line 120-125) were changed to be more concise.

4) Is the SDG flow on during nebulizer characterization, and is the nebulizer flow on during SDG only characterization? L88: Is the flow rate specified otherwise somewhere? It seems redundant.
   The nebulizer is off during SDG characterization. A sentence stating this was added (line 122-123).

5) Section 3.1.1: It's mentioned later, but already here mention that collection by diffusion will overrepresent small particles. Validity of max. primary particle size and Df of small agglomerates is probably unaffected, but the consideration should be part of the authors

analysis. Differences between the metals are consistent with prior literature on spark discharge, not necessarily new.

A sentence stating the overrepresentation was added in the section (line 179-180). Furthermore another sentence was added to clarify that the presented results are also observed by others (Feng et al., 2016; Grammatikopoulos et al., 2014; José-Yacamán et al., 2005; Debecker et al., 2024; Tabrizi et al., 2010), line 222-224 and line 230-231 respectively).

6) L151 "Fully coalesced particles are defined here as primary particles." Is this the same as singlet particles (e.g. Feng 2016) ? The circularity criterion for singlet particle identification is a nice addition to previous analyses, and could be more explicit.

Yes, the definition of the spherical particle size is the same as 'spherical singlet particles' defined by Feng and coworkers. The 'primary particle size' as defined here was also used in (Tabrizi et al., 2009) – we now included this reference in the manuscript (line 187-188). Additionally, one sentence is added in line 192-193 highlighting the suitability of using the sphericity as a parameter to quantify primary particle size limits.

7. The comparison of electrode mass loss and NP mass on filter is very qualitative. It shouldn't be too difficult to get a first order approximation of expected relative mass rates based on the Llewellyn Jones formula (see Tabrizi 2009, Feng 2016), and confirm whether or not the suggested explanations fit.

We thanks the reviewer for this comment and implemented the energy balance equation by Llewellyn Jones (Jones, 1950) for our system and electrodes and added the results in the corresponding section (line 256-261) as well as Table 2.

8. L193: Charge related losses are significant, but not necessarily majority of losses. Losses due to turbulence / poor flow conditions typically are also significant. Collection efficiency on membrane filters is loading dependent, which causes an underestimation of mass arriving at the filter. This effect is most important for low loadings.

We included in the revised manuscript the additional loss processes mentioned in the comment above in line 237-242.

9. L228-235, Fig 4. Are the concentrations for the TEM samples calculated by the diffusion correction, or only the relative abundance?

Yes, we determine particle concentrations for Fig. 4 from the TEM samples taking into account diffusional processes. We now describe the conversion of size distributions derived from TEM analysis to aerosol concentrations in more detail in the manuscript (line 287-290).

10. L281: please provide clear TEM images in SI for the contact angle measurements.

The angle was added in the main figure (Fig. 5) and shown in a more detailed way also in the appendix in Figure A2.

11. Fig 6: Shouldn't this be compared to the observed TEM size distribution in figure 3? The reason why the TEM grids collect more of the smaller particles is the same reason why the TiO2 collect more of the smaller particles.

We now include a comparison of Fig. 6 and Figs. 2 and 3 in the manuscript (line 354-357).

12. Fig 8. EDX is difficult to read, in particular the Au sputtering layer mentioned in L333-335. This type of information must be clearly visible in the graph, e.g. using labels in the graph.
Arrows were added to Fig. 8 (Ni) to make the distinction between Ni particles and Au sputtering layer clearer.

**Minor details:**

13. L84: The set voltage for the spark generator used is the mean voltage, not the breakdown voltage.
The term 'breakdown voltage' was changed to 'discharge voltage' in line 105.

14. There is no forward reference to appendix C. If it's not relevant to the work itself, best to include this in the work referred to in L380.
A forward reference to appendix C was added in line 272.

**Literature:**

Debecker, D. P., Hongmanorom, P., Pfeiffer, T. V., Zijlstra, B., Zhao, Y., Casale, S., and Sassoye, C.: Spark ablation: a dry, physical, and continuous method to prepare powdery metal nanoparticle-based catalysts, Chem. Commun., 60, 11076–11079, https://doi.org/10.1039/D4CC03469D, 2024.

Feng, J., Huang, L., Ludvigsson, L., Messing, M. E., Maisser, A., Biskos, G., and Schmidt-Ott, A.: General Approach to the Evolution of Singlet Nanoparticles from a Rapidly Quenched Point Source, J. Phys. Chem. C, 120, 621–630, https://doi.org/10.1021/acs.jpcc.5b06503, 2016.

Grammatikopoulos, P., Cassidy, C., Singh, V., and Sowwan, M.: Coalescence-induced crystallisation wave in Pd nanoparticles, Sci Rep, 4, 5779, https://doi.org/10.1038/srep05779, 2014.

Jones, F. L.: Electrode Erosion by Spark Discharges, Br. J. Appl. Phys., 1, 60, https://doi.org/10.1088/0508-3443/1/3/302, 1950.

José-Yacamán, M., Gutierrez-Wing, C., Miki, M., Yang, D.-Q., Piyakis, K. N., and Sacher, E.: Surface Diffusion and Coalescence of Mobile Metal Nanoparticles, J. Phys. Chem. B, 109, 9703–9711, https://doi.org/10.1021/jp0509459, 2005.

Tabrizi, N. S., Ullmann, M., Vons, V. A., Lafont, U., and Schmidt-Ott, A.: Generation of nanoparticles by spark discharge, J Nanopart Res, 11, 315–332, https://doi.org/10.1007/s11051-008-9407-y, 2009.

Tabrizi, N. S., Xu, Q., Van Der Pers, N. M., and Schmidt-Ott, A.: Generation of mixed metallic nanoparticles from immiscible metals by spark discharge, J Nanopart Res, 12, 247–259, https://doi.org/10.1007/s11051-009-9603-4, 2010.

---

## Author Comment (AC3)

Answers to comments of reviewer 2

**Comment from the authors to this review:**

Below are given detailed answers to all specific comments and questions of reviewer 2. Most comments by the reviewer have already been covered in the original manuscript (see details below), indicating that the reviewer did not seriously read our manuscript.

A main generic accusation of this review claims that we insufficiently reference previous work. There is always the possibility of missing a relevant publication. However, we reference 67 relevant publications in our manuscript related to all relevant results and discussions in our study. If relevant publications are missing, a proper review should mention them explicitly rather than accusing authors of omitting them intentionally.

We also like to state very clearly that the language and generic accusations in this review are unacceptable for a serious scientific discussion (see also the Chief Editor Comment to this review)! Such reviews are the exact opposite of what a thorough and fair scientific discussion should be, which is the very essence of the entire peer-review process and one of the core foundations of scientific progress. We therefore hope that this review and its style of comments will serve as a pointed negative example of poor scientific discussion and reviewing process.

**General comments**

Overall the work is written in a vague manner without direct means for relevant confirmations. This work also made the impression for reproducing already published works published years ago but not yet fully cited them. This "gave the chance" of this work to pretend to be the initial discoveries. The morphologies of the NPs demonstrate that this study did not yet optimize the system. Only switching on the generator and collecting them for subsequent analysis cannot add scientific values, particularly ignoring the large volume of literature, which reported almost the same line of argumentations. The authors also try to use artificially texted letters to mark the different metals without showing any experimental proofs. Raising such concern is mainly due to all their similar appearances but undistinguishable nature from the TEM results.

See comments above.

1) Primary particle was defined scientifically incorrect.
   We defined Primary Particles as is usually done in similar publications in line 187-188 (see, e.g., (Tabrizi et al., 2009) p.319)

2) Where the authors showed the 1-nm particles?
   The 1 nm particles are visible on the TEM images of Figure 2 and quantified in Figure 3.

3) Melting point of the 4 metals cannot be directly used to evaluate their different particle sizes, as the impure surface of the particles also makes dramatic influences.
   The impure (mostly oxidized) particle surfaces are described in the manuscript already, affecting mostly Cu and Ni. Surface oxidation increases melting points compared to the pure metal (line 214-217). Two more publications were added in line 219 (Seipenbusch et al., 2003) and line 217 (Olszok et al., 2023) describing the oxide formation and limitations in

4) How the authors prove that the oxidation took place during NP production?
   Oxidation during spark production is a process well described in literature using the same conditions as we do; e.g. a gas purity of 5.0. The relevant literature is mentioned and cited in line 215-217. One more paper was added (Olszok et al., 2023).

5) Why the mass was evacuated using micro molar? As the authors also claimed that the SDG can also deliver NPs of high quantity. How the amount of micro molar supports the aforementioned claim?
   The mass was given in micromolar to be able to compare all four metals more easily. Providing mass in milligrams only would complicate this comparison given the different densities of the various metals. This is also done by multiple other studies such as (Schmidt-Ott, 2019; Tabrizi et al., 2009) The term 'high quantity' is used to define the *number* of particles (in the range of 1E8 particles per cm$^3$).

6) For the different production rates of the metals, previous study already clearly explained why. Please cite the relevant literature, not only saving the unnecessarily repeated work but also not making an impression of confined literature review.
   The paragraph describing the different production rates was extended to include more than one literature reference (Domaschke et al., 2018; Schmidt-Ott, 2019). L254-261

7) The authors also argued that the mismatch of size distributions measured between the TEM analysis and SMPS data is mainly due to the low counting efficiency from the latter. How to explain the overestimation of diffusional deposition of smaller NPs for TEM analysis?
   The overestimation of smaller particles via the deposition through diffusion is described in the manuscript (line 281-290). Therein, we also discuss in detail how to correct this collection bias and show results of this correction in Figure 4.

8) A very puzzled presentation for Fig. 5 onwards is to remove the Cu NPs. Could the authors provide any scientific reasons for such inconsistencies?
   The reason why the results for Cu were not considered anymore in the main text of the manuscript can be found in line 324-325 A sentence was added on where the data on Cu can be found (i.e. appendix B) in line 325.

9) How the edges of the NPs were determined when using eq. 1 to calculate circularity?
   This is described in detail in line 141-145.

10) How the authors separate the ones coated over TiO2 with those of self-coagulation?
    This topic is thoroughly discussed in the manuscript in line 361-379.

**Literature**

Domaschke, M., Schmidt, M., and Peukert, W.: A model for the particle mass yield in the aerosol synthesis of ultrafine monometallic nanoparticles by spark ablation, Journal of Aerosol Science, 126, 133–142, https://doi.org/10.1016/j.jaerosci.2018.09.004, 2018.

Olszok, V., Bierwirth, M., and Weber, A. P.: Creation of Gases with Interplanetary Oxygen Concentration at Atmospheric Pressure by Nanoparticle Aerosol Scavengers: Implications for Metal Processing from nm to mm Range, ACS Appl. Nano Mater., 6, 1660–1666, https://doi.org/10.1021/acsanm.2c04585, 2023.

Schmidt-Ott, A. (Ed.): Spark Ablation: Building Blocks for Nanotechnology, Jenny Stanford Publishing, New York, 472 pp., https://doi.org/10.1201/9780367817091, 2019.

Seipenbusch, M., Weber, A. P., Schiel, A., and Kasper, G.: Influence of the gas atmosphere on restructuring and sintering kinetics of nickel and platinum aerosol nanoparticle agglomerates, Journal of Aerosol Science, 34, 1699–1709, https://doi.org/10.1016/S0021-8502(03)00355-0, 2003.

Tabrizi, N. S., Ullmann, M., Vons, V. A., Lafont, U., and Schmidt-Ott, A.: Generation of nanoparticles by spark discharge, J Nanopart Res, 11, 315–332, https://doi.org/10.1007/s11051-008-9407-y, 2009.

---

## Author Response (AR2)

**Answers to comments of reviewer 1**

The revision has significantly improved the MS. However, there are still some points that need to be clarified or corrected. They are listed below. As the MS contains many valuable results and findings, it would be a pity not to publish it. But major revisions are necessary first.

The authors like to thank the reviewer again for their constructive comments. Our answers to their questions and comments are given below in blue font.

- There are still some inconsistencies within the MS that should be eliminated by careful revision, e.g. in the abstract, the TiO2 NP size is given as 120 nm while on page 3 it is stated as 100 nm. Please be consistent.
  We changed the 100 to 120 nm in the manuscript (L 79) and checked the whole text again for any other inconsistencies.
- 2) Please be clear with the flow rates: 5.8 lpm for total flow, divided into 4.6 lpm (SDG) and 1.2 lpm (nebulizer). In this context, why was the flow rate through the SDG not kept constant? Does this not change the PSD of the metal NPs? Would it not have been better, in the case without TiO2 NP, to run the nebulizer empty and keeping the flow rate in the SDG constant at 4.6 lpm?

We added another sentence in L 111-112 and changed an existing sentence to be more clear about the flow rates (L 116-117). Running the nebuliser empty would have been possible. It is true that the PSD might be affected by the flow changes in our set up. However, the change in the PSD between 4.6 l/min and 5.8 l/min is small enough to be negligible. See for example (Tabrizi et al., 2009). A sentence stating this was also added in L 118-119.

 While the "adsorption" of gas molecules is common for instance in catalysis or BET measuremnts, the term "adsorbed" metallic NPs is rather unusual. Please use "attached" for particles.

We changed this as suggested.

4) In table 2 and the corresponding discussion there is something wrong. First, where do the electrical conductivities come from? According to the Wiedemann-Franz law, which applies peculiarly well for pure metals, the ratio of thermal to electrical conductivity should be nearly constant. But when calculating these ratios completely different values are obtained. In fact, what is shown in the "electric conductivity" column should be the "electrical conductance" in S/m and not in Ohm\*m. The true values for Au and Cu are 45 E6 S/m and 58 E6 S/m, respectively. As a consequence also the argument of Joule heating does not apply specifically to gold. As stated in Tabrizi et al. (2009) the liquid pool which forms on the electrode for a short time during discharge is a local phenomenon as is the ejection of metal droplets ("jet droplets") when the pressure due to ion bombardment ceases. There must be another reason for the higher release rate of Au and particularly for the lower release rate of Cu which is not included in all the available models (Jones, Tabrizi, Pfeiffer, Domaschke,...). We thank you for pointing this out. We falsely listed the electrical resistivity instead of the electrical conductivity. We changed Table 2 accordingly and do not list either of these

properties because the electrical conductivity does not correlate with our observed particle. We excluded the discussion about the liquid pools given the changed electrical conductivity values. As you rightly said, there must be another reason for the higher release rate/losses of Au. We added a discussion in the paper on L 240-242 connecting the electrode vapor density to the particle formation and growth rate and thus also on the diffusion loss rate in the spark chamber: The more ablation (i.e. mass loss), the higher the electrode vapor density, the faster the particle growth and the lower the diffusion losses in the spark chamber.

We also added a sentence in L 233-236 highlighting the influence of the thermal conductivity on the mass loss and added a reference (Loizidis et al., 2024) to underline the statement. Furthermore, there was mix up in the bulk melting temperatures for Cu and Ni in Table 2, this was fixed also.

5) The range of the fractal agglomerates remains unclear. In Table 1 there are values for Df obtained by nested circles method. However, over which size range was this analysis performed. On page 13 it is stated that accurate size characerization is increasingly difficult for larger agglomerates. But if SMPS measurements were done such a size is known (nearly the projection equivalent diameter) and also if the TEM micrographs have been evaluated to determine Df such a collision diameter can be calculated (e.g. as radius of gyration). Please indicate in Table 1 the diameter range of applicability of Df.

The column header about the fractal dimension in Table 1 now describes in more detail the size range for which the fractal dimensions were determined. They were determined for particles being larger than the maximal spherical (i.e. primary) particle diameter (i.e. for all agglomerates). Furthermore, a sentence on page 13 (line 295-297) was changed to be clearer.

- 6) In Fig. 5 and in the appendix some examples of contact angles are presented. However, the quality of the TEM micrographs is not that high. On a statistical ground, how many contact angles have been evaluated to obtain the given values and how large is the uncertainty? We now evaluated the contact angles of 5 particles. Unfortunately, our current TEM analyses do not allow to determine the contact angles of more particles. We adjusted the values of the angles and added the standard deviations on line 325-326 and in the appendix (Fig. A2).
- 7) The data evaluation to obtain Fig. 7 is still unclear. If I understood right, all determinations are based on the counting of metallic NPs on the TEM grid, once without TiO2 NPs and once with TiO2 NPs. Then, the legend in Fig. 7 is misleading. In coatings studies X@Y is used to indicate that the coating material X is attached to the core material Y. However, here it is ment that the free metallic NPs on the TEM gris are counted in the presence of TiO2 NPs. Please use another descriptor for this situation. In addition, when looking at Fig. C1(a) in the Appendix, nearly all metallic NPs should be attached to the TiO2 NPs after 26 s. Why is this residence time (26 s) not included in Fig. 7 as a third diagram besides 1.3 s and 2.2 s? This would make it immediately clear what the meaning of Fig. 7 is.

We renamed the descriptor for the uncoated metal particles on the TEM grids during coating with  $TiO_2$  as: 'Metal during coating' and the case without  $TiO_2$  present as 'Metal only' in Figs. 7 and A4. The 26 s residence time is omitted given the large uncertainties in the conversion to an aerosol number size distribution due to the small number of particles found on the TEM grids at this residence time also when no  $TiO_2$  was present (c.f. Fig. 3 inlays for Au and Pt). In both cases, with or without coating, no metal number size distributions could be determined. So even though the coating results in almost no uncoated remaining metal particles, this cannot be shown with the way of analysis shown in Fig. 7.

8) In the conclusions, it is stated that "... could be related to their thermodynamic properties such as melting point or surface free energy." However, the influence of the surface free energy was never treated nor mentioned before. Please be consistent, either by introducing and discussing the surface free energy earlier in the MS or by removing the term in the conclusions.

We changed the sentence in the conclusion of the manuscript (L 389-390) now not mentioning the surface free energy anymore. We also deleted the mentioning of the surface free energy from the abstract (L 16-17).

**Literature**

Loizidis, C., Petallidou, K. C., Maisser, A., Bezantakos, S., Pfeiffer, T. V., Schmidt-Ott, A., and Biskos, G.: Insights into the enhancement of nanoparticle production throughput by atmospheric-pressure spark ablation, Aerosol Science and Technology, 58, 1421–1431, https://doi.org/10.1080/02786826.2024.2403578, 2024.

Tabrizi, N. S., Ullmann, M., Vons, V. A., Lafont, U., and Schmidt-Ott, A.: Generation of nanoparticles by spark discharge, J Nanopart Res, 11, 315–332, https://doi.org/10.1007/s11051-008-9407-y, 2009.